# Quantile Multi-Armed Bandits with 1-bit Feedback

**Ivan Lau**        IVAN.LAU@U.NUS.EDU  and  **Jonathan Scarlett**        SCARLETT@COMP.NUS.EDU.SG
*National University of Singapore*

**Editors:** Gautam Kamath and Po-Ling Loh

## Abstract

In this paper, we study a variant of best-arm identification involving elements of risk sensitivity and communication constraints. Specifically, the goal of the learner is to identify the arm with the highest quantile reward, while the communication from an agent (who observes rewards) and the learner (who chooses actions) is restricted to only one bit of feedback per arm pull. We propose an algorithm that utilizes noisy binary search as a subroutine, allowing the learner to estimate quantile rewards through 1-bit feedback. We derive an instance-dependent upper bound on the sample complexity of our algorithm and provide an algorithm-independent lower bound for specific instances, with the two matching to within logarithmic factors under mild conditions, or even to within constant factors in certain low error probability scaling regimes. The lower bound is applicable even in the absence of communication constraints, and thus we conclude that restricting to 1-bit feedback has a minimal impact on the scaling of the sample complexity.

**Keywords:** Best-Arm identification, quantile bandits, 1-bit quantization

## 1. Introduction

The multi-armed bandit (MAB) is a well-studied decision-making framework due to its effectiveness in modelling a wide range of application domains such as online advertising, recommendation systems, clinical trials, and A/B testing. Two common but distinct objectives in theoretical MAB studies are regret minimization and best arm identification (BAI), and this paper is focused on the latter. The goal of BAI is for the learner/decision-maker to efficiently identify the "best" arm (decision) from a set of arms, where the learning process occurs through "pulling" the arms and receiving some feedback about their rewards.

In the vanilla setting of BAI, the best arm is defined as the arm whose reward distribution has the highest mean, and the learner has access to direct observations of the rewards of the pulled arms. To tailor to certain practical applications, the best arm is sometimes defined using a different performance measure, and certain constraints are sometimes incorporated into the feedback/learning process. Examples of this include (but are not limited to) the following:

(i)  in settings where the decision-making is risk-sensitive, using quantiles or value-at-risk as the performance measure may be more appropriate than using mean reward;

(ii) in settings where the uplink communication from the sensor to the server (learner) is costly, the communication to the learner may be restricted, e.g., to send only a few bits rather than sending the exact reward.

In this paper, we consider a setup for BAI that features both of these aspects. Specifically, the communication to the learner is restricted to *one bit* of feedback per arm pull, and the goal of the learner is to identify the arm with the highest $q$-quantile for some $q \in (0, 1)$. The problem setup is described formally in Section 2.1. The main contribution of this paper is an algorithm (Algorithm 1) for this problem whose upper bound on the sample complexity nearly matches the lower bound for the problem *without* the communication constraint. This complements an analogous study of highest mean BAI giving evidence that *multiple bits per arm pull* need to be used (Hanna et al., 2022), suggesting that the quantile-based objective may be easier to handle in highly quantized scenarios. The details of our results and contributions are given in Section 2.2. Before formally introducing the problem and stating our contributions, we outline some related work.

## 1.1. Related Work

The multi-armed bandit (MAB) problem was first studied in the context of clinical trials in (Thompson, 1933) and was formalized as a statistical problem in (Robbins, 1952). The related work on MAB is extensive (e.g., see (Slivkins, 2019; Lattimore and Szepesvári, 2020) and the references therein); we only provide a brief outline here, emphasizing the most closely related works.

**Best arm identification.** The early work on MAB focused on balancing the trade-off between exploration and exploitation for cumulative regret minimization. The best arm identification (BAI) problem was introduced in (Even-Dar et al., 2002) as a "pure exploration" problem, where the goal is to find from an arm set $\mathcal{A}$, the arm $k^* = \text{argmax}_{k \in \mathcal{A}} \mu_k$ with the highest mean reward (the "best" arm). Subsequent work on BAI includes (Bubeck et al., 2009; Audibert and Bubeck, 2010; Gabillon et al., 2012; Karnin et al., 2013; Jamieson et al., 2014; Kaufmann et al., 2016; Garivier and Kaufmann, 2016), and these are commonly categorized into the fixed budget setting and fixed confidence setting. In the fixed confidence setting, which is the focus of our work, the target error probability is fixed, and the objective is to devise an algorithm that identifies the best arm, in the Probably Approximately Correct (PAC) sense, using a minimal average number of arm pulls. Formally, an algorithm is $\delta$-PAC correct if it satisfies $\sup_\nu \mathbb{P}(\hat{k} \neq k^*) \leqslant \delta$, where $\hat{k}$ is the output of the algorithm, $k^*$ is the best arm, and the supremum is taken over the collection of instances $\nu$ such that there exists a unique best arm. A lower bound of $\sum_{k \neq k^*} \Delta_k^{-2} \log(\delta^{-1})$ on the sample complexity was given in (Mannor and Tsitsiklis, 2004), where $\Delta_k = \mu_{k^*} - \mu_k$ is the arm suboptimiality gap. Several subsequent algorithms managed to match the dependence on $\Delta_k$ of the lower bound to within a doubly logarithmic factor. Despite the multitude of algorithms, these are usually based on one of the following general sampling strategies: arm/action elimination, upper confidence bounds (UCB), lower upper confidence bound (LUCB), and Thompson sampling. See (Jamieson and Nowak, 2014) for an overview and relationships between these sampling strategies.

**Quantile bandits.** In certain real-world applications, the mean reward does not satisfactorily capture the merits of certain decisions. This has motivated the use of other risk-aware performance measures in place of the mean (Yu and Nikolova, 2013), such as the mean-variance risk, the (conditional)

value-at-risk, and quantile rewards – see (Tan et al., 2022) for an extensive survey. Among these, our work is most closely related to the quantile multi-armed bandit problem (QMAB), a variant of the MAB problem in which the learner is interested in the arm(s) with the highest quantile reward (e.g., the median). This is useful when dealing with heavy-tailed reward distributions or risk-sensitive applications, where a decision-maker might prioritize minimizing risk by focusing on lower quantiles (e.g., optimizing worst-case outcomes) or targeting the top-performing outcomes by focusing on higher quantiles. In particular, (Szorenyi et al., 2015; David and Shimkin, 2016; Nikolakakis et al., 2021; Howard and Ramdas, 2022) studied QMAB in BAI in the fixed confidence setting. Compared to mean-based BAI, the definition of the arm suboptimality gap $\Delta_k$ is not as straightforward, but this has been resolved in (Nikolakakis et al., 2021; Howard and Ramdas, 2022). Based on the suboptimiality gap, a lower bound of the form $\sum_k \Delta_k^{-2} \log(\delta^{-1})$ was given in (Nikolakakis et al., 2021) for suitably-defined $\Delta_k$. Algorithms in (Nikolakakis et al., 2021) and (Howard and Ramdas, 2022), which are based on arm elimination and LUCB respectively, were shown to match the dependence on $\Delta_k$ of the lower bound, to within a doubly logarithmic factor. Other variants of quantile bandit problems include fixed confidence median BAI with contaminated distributions (Altschuler et al., 2019); fixed confidence quantile BAI with differential privacy (Nikolakakis et al., 2021); fixed budget quantile BAI (Zhang and Ong, 2021); and quantile bandit regret minimization (Torossian et al., 2019).

**Communication-constrained bandits.** Most work in MAB assumes that the arms' reward can be observed directly by the learner (with full precision). However, this assumption may be impractical for real-world applications in which the reward observations are done by some agent (sensor) before being communicated to the learner (central server). This motivated the distributed MAB framework, which has garnered significant attention in recent research; see (Amani et al., 2023), (Salgia and Zhao, 2023, Appendix A) and the references therein. The distributed MAB studies most pertinent to this work are those that focused on the quantization of the reward feedback communicated from agent to learner (Vial et al., 2020; Hanna et al., 2022; Mitra et al., 2023; Mayekar et al., 2023), which is motivated by applications where uplink communication bandwidth is limited (e.g., those using low-power sensors such as drones and wearable healthcare devices). In particular, (Vial et al., 2020; Hanna et al., 2022) studied constant bit quantization schemes for cumulative regret minimization problem in mean-based bandits, where only a constant number of bits are used to communicate each reward observation. They showed that if the rewards are all supported on $[0, 1]$, then there exists a 1-bit quantization scheme that can achieve regret comparable to those in unquantized setups. However, (Hanna et al., 2022, Sec. 3) showed that if the rewards are supported on $[0, \lambda]$ for general $\lambda > 0$, then the same scheme would result in a regret that scales linearly in $\lambda$. They further established that, in order to attain a natural set of sufficient (albeit not necessary) conditions for matching the unquantized regret to within a constant factor, at least 2.2 bits per reward observation are necessary. This suggests a possible inherent challenge, or at least a need for different techniques, when using 1-bit quantization. Finally, while some distributed MAB studies considered BAI problems (Hillel et al., 2013; Karnin et al., 2013; Tao et al., 2019; Réda et al., 2022), we are unaware of any that addressed the number of bits of feedback per round or used quantile-based performance measures.

## 2. Problem Setup and Contributions

### 2.1. Problem Setup

We study the following variant of fixed-confidence best arm identification for quantile bandits.

**Arms and quantile rewards.** The learner is given a set of arms $\mathcal{A} = \{1, 2, \ldots, K\}$ with a stochastic reward setting. That is, for each arm $k \in \mathcal{A}$, the observations/realizations of its reward are i.i.d. random variables from some fixed but unknown reward distribution with CDF $F_k$. This defines a (lower) quantile function $Q_k \colon [0, 1] \to \mathbb{R}$ for each $k \in \mathcal{A}$ as follows:[1]

$$Q_k(p) := \sup\{x \in \mathbb{R} : F_k(x) < p\} = \inf\{x \in \mathbb{R} : F_k(x) \geqslant p\}. \tag{1}$$

The learner is interested in identifying an arm $\hat{k}$ with the highest $q$-quantile. While the reward of each arm is allowed to be unbounded, we assume the $q$-quantile of each arm to be bounded in a known range $[0, \lambda]$.[2] We let $\mathcal{P} = \mathcal{P}(q, \lambda)$ denote the collection of all distributions with $q$-quantile in $[0, \lambda]$, and let $\mathcal{E} := \mathcal{P}^K$ be the collection of all possible instances the learner could face. We will sometimes write $\mathbb{P}_\nu[\cdot]$ and $\mathbb{E}_\nu[\cdot]$ to explicitly denote probabilities and expectations under an instance $\nu \in \mathcal{E}$.

**1-bit communication constraint.** We frame the problem as having a single learner that makes decisions, and a single agent that observes rewards and sends information on them to the learner. In Remark 1 below, we discuss how this can also have a multi-agent interpretation. With a single agent, the following occurs at each iteration/time $t \geqslant 1$ indexing the number of arm pulls:

1. The learner asks the agent to pull an arm $a_t \in \mathcal{A}$, and sends the agent some side information $S_t$.

2. The agent pulls $a_t$ and observes a random reward $r_{a_t, t}$ distributed according to CDF $F_{a_t}$.

3. The agent transmits a 1-bit message to the learner, where the message is based on $r_{a_t, t}$ and $S_t$.

4. The learner decides on arm $a_{t+1} \in \mathcal{A}$ and side information $S_{t+1}$, based on arms and the 1-bit information received in iterations $1, \ldots, t$.

We will focus on the *threshold query model*, where at iteration $t$, the side information $S_t$ is a query of the form "Is $r_{a_t, t} \leqslant \gamma_t$?" and the 1-bit message is the corresponding binary feedback $\mathbf{1}\{r_{a_t, t} \leqslant \gamma_t\}$. The learner will only use such queries as side information in our algorithm, though the problem itself is of interest for both threshold queries and general 1-bit quantization methods (possibly having different forms of side information).

**Remark 1** *We do not impose any (downlink) communication constraint from the learner to the agent, as this cost is typically not expensive. While we framed the problem as having a single agent for clarity, we are motivated by settings where the agent at each time instant could potentially correspond to a different user/device. For this reason, and also motivated by settings where agents are low-memory sensors, we assume that the agent is 'memoryless', meaning the 1-bit message transmitted cannot be dependent on rewards observed from previous arm pulls. The preceding*

---

1. The equality follows from the right-continuity of $F_k$.

2. We note that setting the lower limit to 0 is without loss of generality, and regarding the interval length $\lambda$, even a crude upper bound is reasonable since the sample complexity will only have logarithmic dependence; see Theorem 14.

*assumptions were similarly adopted in some of the most related previous works (Hanna et al., 2022; Mitra et al., 2023; Mayekar et al., 2023).*

$\epsilon$-**relaxation.** Fix a QMAB instance $\nu \in \mathcal{E}$, and let $k^* \in \mathcal{A}$ be an arm with the largest $q$-quantile for the instance $\nu$. Instead of insisting on identifying an arm with the exact highest quantile, we relax the task by only requiring the identified arm $\hat{k}$ to be at most $\epsilon$-suboptimal in the following sense:

$$\hat{k} \in \mathcal{A}_\epsilon(\nu) := \left\{ k \in \mathcal{A} \mid Q_k(q) \geqslant Q_{k^*}(q) - \epsilon \right\}. \tag{2}$$

This allows us to limit the effort on distinguishing arms whose $q$-quantile rewards are very close to each other; analogous relaxations are common in the BAI literature. This relaxation is also motivated by the threshold query model mentioned above; specifically, we will see in Section 4.2 that achieving (2) under the threshold query model requires $\Omega(\log(\lambda/\epsilon))$ arm pulls even in the case of *deterministic* two-arm bandits. Our goal is to design an algorithm to identify an arm satisfying (2) with high probability while using as few arm pulls as possible.

## 2.2. Summary of Contributions.

With the problem setup now in place, we summarize our main contributions as follows:

- We provide an algorithm (Algorithm 1) for our setup, with the uplink communication satisfying the 1-bit constraint. Unlike standard bandit algorithms that compute empirical statistics using lossless observations of rewards, we use a noisy binary search subroutine for the learner to estimate the quantile rewards (see Appendix A).

- We introduce fundamental arm gaps $\Delta_k$ (Definition 10) that generalize those proposed in prior work (see Remark 12). These gaps capture the difficulty of our problem setup in the sense that the problems with positive gaps essentially coincide with the set of problems that are solvable; see Theorem 21 and Remark 23 for precise statements.

- We provide an instance-dependent upper bound on the number of arm pulls to guarantee (2) with high probability (Corollary 15), expressed in terms of $\lambda$, $\epsilon$, and fundamental arm gap $\Delta_k$. Our upper bound scales logarithmically with $\lambda/\epsilon$, which contrasts with the existing upper bound for mean-based bandits with 1-bit quantization scaling linearly with $\lambda$ (Vial et al., 2020; Hanna et al., 2022).

- We also derive a worst-case lower bound (Theorem 16) showing that our upper bound is tight to within logarithmic factors under mild conditions, and can even be tight to within constant factors when the target error probability $\delta$ decays to zero fast enough. We additionally provide a lower bound (Theorem 17) showing that $\Omega(\log(\lambda/\epsilon))$ dependence is unavoidable under threshold queries in arbitrary scaling regimes. The former lower bound is applicable even in the absence of communication constraints, so we can conclude that restricting to 1-bit feedback has a minimal impact on the sample complexity, at least in terms of scaling laws.

## 3. Algorithm and Upper Bound

In this section, we introduce our main algorithm and provide its performance guarantee.

### 3.1. Description of the Algorithm

Our algorithm (Algorithm 1) is based on successive elimination, which is well-studied in the standard BAI problem and has also been adapted for other variations. The algorithm pulls arms in *rounds*, where each round consists of multiple pulls (namely, pulling all non-eliminated arms). For each arm $k$ that is active at round $t$,[3] the algorithm computes a confidence interval $[\mathrm{LCB}_t(k), \mathrm{UCB}_t(k)]$ that contains the $q$-quantile $Q_k(q)$ with high probability (see Lines 12 and 14). Based on the confidence intervals, the algorithm eliminates arms that are suboptimal (see Line 15). When the algorithm identifies that some arm satisfies (2) based on the confidence bounds, it terminates and returns that arm (see Lines 8 and 17).

This high-level idea was also used in (Szorenyi et al., 2015; Nikolakakis et al., 2021) for the quantile bandit problem with no communication constraint, but the procedures to obtain the confidence intervals are very different. Their confidence intervals are computed using empirical quantiles of the (direct) observed rewards, which our learner does not have the luxury of accessing. Instead, we discretize the continuous interval $[0, \lambda]$ to a discrete interval $[0, \tilde{\epsilon}, 2\tilde{\epsilon}, \ldots, \lambda]$,[4] and use a quantile estimation algorithm QuantEst to help us find $\mathrm{LCB}_t(k)$ and $\mathrm{UCB}_t(k)$ from the discretized interval; see Lines 1–2 and Lines 11–14 respectively. QuantEst can be implemented in our problem setup while respecting the 1-bit uplink communication constraint: the learner sends threshold queries in the form "Is $r_{a_t,t} \leqslant \gamma_t$?" to the agent and receives 1-bit comparison feedback $\mathbf{1}(r_{a_t,t} \leqslant \gamma_t)$. Based on the feedback received, the learner then uses a noisy binary search strategy to compute $\mathrm{LCB}_t(k)$ and $\mathrm{UCB}_t(k)$. The details of QuantEst are deferred to Algorithm 2 in Appendix A. For now, we only need to treat QuantEst as a "black box" with the following guarantee: Given input CDF $F$, non-decreasing list $\mathbf{x} = [x_1, \ldots, x_n]$, quantile of interest $\tau \in (0, 1)$, approximation parameter $\Delta \leqslant \min(\tau, 1 - \tau)$ and probability parameter $\delta$, QuantEst$(F, \mathbf{x}, \tau, \Delta, \delta)$ will use at most $O\left(\frac{1}{\Delta^2} \log \frac{n}{\delta}\right)$ threshold queries and output an index $i$ satisfying $\mathbb{P}\left([F(x_i), F(x_{i+1})] \cap (\tau - \Delta, \tau + \Delta) = \varnothing\right) < \delta$. Formally, the guarantees on its outputs $l_{t,k}$ and $u_{t,k}$ (see Lines 11 and 13) as well as the number of arm pulls used are stated as follows.

**Lemma 2 (Good event)** *Fix an instance $\nu \in \mathcal{E}$, and suppose Algorithm 1 is run with input $(\mathcal{A}, \lambda, \epsilon, q, \delta)$ and parameter $c \geqslant 1$. Let $\Delta^{(t)}$, $\mathcal{A}_t$, $l_{t,k}$, $u_{t,k}$ be as defined in Algorithm 1 for each*

---

3. We slightly abuse notation and use $t$ to index "rounds", each consisting of several arm pulls; it will be clear from the context whether $t$ is indexing a round or indexing the number of pulls so far. We still use the *total* number of arm pulls to characterize the performance of the algorithm.

4. We use an input parameter $c > 0$ to control how finely the continuous interval is discretized; see Remark 3.

5. The distance between $x_i$ and $x_{i+1}$ for $1 \leqslant i \leqslant n$ is exactly $\tilde{\epsilon}$, which is approximately $\epsilon/(c+1)$ (up to the impact of rounding in Line 1). We choose the spacings to be equal for ease of analysis.

6. We add $\pm\infty$ to the ends of the list $\mathbf{x}$ to handle the edge cases $F_k(0) = q$ and $F_k(\lambda) = q$. Without this, Lemma 2 may not be satisfied: $[F_k(0), F_k(\tilde{\epsilon})] \cap (q - \Delta^{(t)}, q) = \varnothing$ and $[F_k(\lambda - \tilde{\epsilon}), F_k(\lambda)] \cap (q, q + \Delta^{(t)}) = \varnothing$.

7. We use the convention that the maximum of an empty set is $-\infty$, so the while-loop termination condition is trivially satisfied when $|\mathcal{A}_t| = 1$.

---

**Algorithm 1** Main Algorithm

---

**Input**: Arms $\mathcal{A} = \{1, \ldots, K\}$, and $\lambda, \epsilon, q, \delta$, where $\lambda > \epsilon$ and $q, \delta \in (0, 1)$

**Parameter**: $c \in \mathbb{Z}^+$

1: Set $n := \lceil (c+1)\lambda/\epsilon \rceil$
2: Set $\tilde{\epsilon} := \lambda/n$ [5]
3: Initiate a list $\mathbf{x} = [x_0, x_1, \ldots, x_n, x_{n+1}, x_{n+2}] = [-\infty, 0, \tilde{\epsilon}, 2\tilde{\epsilon}, \ldots, (n-1)\tilde{\epsilon}, \lambda, \infty]$ [6]
4: Initiate round index $t = 1$
5: Initiate the set of active arms $\mathcal{A}_t = \mathcal{A} = \{1, \ldots, K\}$
6: **for** arm $k \in \mathcal{A}_t$ **do**
7: $\quad$ $\text{LCB}_0(k) = x_1 = 0$; $\text{UCB}_0(k) = x_{n+1} = \lambda$
8: **while** $\text{LCB}_{t-1}(k) < \max\limits_{a \in \mathcal{A}_t \setminus \{k\}} \text{UCB}_{t-1}(a) - (c+1)\tilde{\epsilon}$ for all arm $k \in \mathcal{A}_t$ **do** [7]
9: $\quad$ $\Delta^{(t)} \leftarrow 2^{-t+1} \cdot \min(q, 1-q)$
10: $\quad$ **for** arm $k \in \mathcal{A}_t$ **do**
11: $\quad\quad$ Run `QuantEst` (Algorithm 2) with input $\left(F_k, \mathbf{x}, q - \frac{\Delta^{(t)}}{2}, \frac{\Delta^{(t)}}{2}, \frac{\delta \cdot \Delta^{(t)}}{2|\mathcal{A}_t|}\right)$ to obtain an index $l_{t,k} \in \{0, \ldots, n+1\}$
12: $\quad\quad$ $\text{LCB}_t(k) = \max\left(x_{l_{t,k}}, \text{LCB}_{t-1}(k)\right)$
13: $\quad\quad$ Run `QuantEst` (Algorithm 2) with input $\left(F_k, \mathbf{x}, q + \frac{\Delta^{(t)}}{2}, \frac{\Delta^{(t)}}{2}, \frac{\delta \cdot \Delta^{(t)}}{2|\mathcal{A}_t|}\right)$ to obtain an index $u_{t,k} \in \{0, \ldots, n+1\}$
14: $\quad\quad$ $\text{UCB}_t(k) = \min\left(x_{u_{t,k}+1}, \text{UCB}_{t-1}(k)\right)$
15: $\quad$ Update $\mathcal{A}_{t+1} = \left\{k \in \mathcal{A}_t : \text{UCB}_t(k) > \max\limits_{a \in \mathcal{A}_t} \text{LCB}_t(a)\right\}$
16: $\quad$ Increment round index $t \leftarrow t + 1$
17: **return** any arm $\hat{k} \in \mathcal{A}_t$ satisfying $\text{LCB}_t(\hat{k}) \geqslant \max\limits_{a \in \mathcal{A}_t \setminus \{\hat{k}\}} \text{UCB}_t(a) - (c+1)\tilde{\epsilon}$

---

*round index $t \geqslant 1$ and each arm $k \in \mathcal{A}_t$. Define events $E_{t,k,l}$ and events $E_{u,k,l}$ respectively by*

$$E_{t,k,l} := \left\{ [F_k(x_{l_{t,k}}), F_k(x_{l_{t,k}+1})] \cap (q - \Delta^{(t)}, q) \text{ is non-empty} \right\} \tag{3}$$

*and*

$$E_{t,k,u} := \left\{ [F_k(x_{u_{t,k}}), F_k(x_{u_{t,k}+1})] \cap (q, q + \Delta^{(t)}) \text{ is non-empty} \right\}. \tag{4}$$

*Then the Event $E$ defined by*

$$E := \bigcap_{t \geqslant 1} \bigcap_{k \in \mathcal{A}_t} (E_{t,k,l} \cap E_{t,k,u}) \tag{5}$$

*occurs with probability at least $1 - \delta$. Furthermore, for each $t$ and $k \in \mathcal{A}_t$, the number of arm pulls used by* `QuantEst` *to output $l_{t,k}$ and $u_{t,k}$ scales as*

$$O\left(\frac{1}{(\Delta^{(t)})^2} \log\left(\frac{2n|\mathcal{A}_t|}{\delta \Delta^{(t)}}\right)\right) = O\left(\frac{1}{(\Delta^{(t)})^2} \cdot \left(\log\left(\frac{1}{\delta}\right) + \log\left(\frac{1}{\Delta^{(t)}}\right) + \log\left(\frac{c\lambda K}{\epsilon}\right)\right)\right), \tag{6}$$

*where $n = \lceil (c+1)\lambda/\epsilon \rceil$ and $\Delta^{(t)} = 2^{-t+1} \cdot \min(q, 1-q)$ as stated in Lines 1 and 9 of Algorithm 1.*

**Proof** See Appendix A for the details, in which we make use of a noisy binary search subroutine from (Gretta and Price, 2024). ∎

**Remark 3** *We note that the parameter $c \geqslant 1$ in the algorithm controls how finely the continuous interval $[0, \lambda]$ is discretized; one can think of $c = 1$ for simplicity to have roughly $n = 2\lambda/\epsilon$ discretization points spaced by roughly $\epsilon/2$, but we will see in Section 5 that picking a larger value of $c$ can be beneficial.*

## 3.2. Anytime Quantile Bounds

Under Event $E$ as defined in Lemma 2, we obtain the following anytime bounds for the quantiles when running Algorithm 1. These bounds will be used in the proofs of the correctness of Algorithm 1 (Theorem 8) and the upper bound on the number of arm pulls (Theorem 14).

**Lemma 4 (Anytime quantile bounds)** *Fix an instance $\nu \in \mathcal{E}$, and suppose Algorithm 1 is run with input $(\mathcal{A}, \lambda, \epsilon, q, \delta)$ and parameter $c \geqslant 1$. Let $\tilde{\epsilon} = \tilde{\epsilon}(\lambda, \epsilon, c)$, and $\Delta^{(t)}$, $\mathcal{A}_t$, $\mathrm{LCB}_t(k)$, and $\mathrm{UCB}_t(k)$ be as defined in Algorithm 1 for each round index $t \geqslant 1$ and each arm $k \in \mathcal{A}_t$. Under Event $E$ as defined in Lemma 2, we have the following bounds for the arms' lower quantile functions $Q_k(\cdot)$ and upper quantile functions $Q_k^+(p) := \sup\{x \mid F_k(x) \leqslant p\}$:*

$$\mathrm{LCB}_\tau(k) \leqslant \mathrm{LCB}_t(k) < Q_k(q) \leqslant Q_k^+(q) \leqslant \mathrm{UCB}_t(k) \leqslant \mathrm{UCB}_\tau(k) \tag{7}$$

$$Q_k^+\big(q - \Delta^{(t)}\big) \leqslant \mathrm{LCB}_t(k) + \tilde{\epsilon} \tag{8}$$

$$\mathrm{UCB}_t(k) < Q_k\big(q + \Delta^{(t)}\big) + \tilde{\epsilon} \tag{9}$$

*for all rounds $t > \tau \geqslant 0$ and each arm $k \in \mathcal{A}_t$.*

**Proof** This follows from applying properties of quantile functions to events $E_{t,k,l}$ and $E_{t,k,u}$ defined in (3) and (4); see Appendix B for the details. ∎

**Remark 5** *The property that $\mathrm{LCB}_t(k)$ is non-decreasing in $t$, i.e., the first inequality of (7), may appear to be enforced "artificially" by Line 12 of Algorithm 1. It will turn out that this property is crucial in proving Lemma 32, which in turn is important for the analysis in upper bounding the number of arm pulls – see Remark 34.*

## 3.3. Correctness

In this section, we give the performance guarantee of Algorithm 1 using the anytime quantile bounds (Lemma 4). We first formalize the notion of an algorithm returning an *incorrect* output with at most a small error probability $\delta$.

**Definition 6 ($(\epsilon, \delta)$-reliable.)** *Consider an algorithm $\pi$ for the QMAB problem with quantized or unquantized rewards that takes $(\mathcal{A}, \lambda, \epsilon, q, \delta)$ as input and operates on instances $\nu \in \mathcal{E}$. Then, we*

say $\pi$ is $(\epsilon, \delta)$-*reliable if for each instance* $\nu \in \mathcal{E}$, *it returns an* incorrect *output with probability at most* $\delta$, *i.e.*,

$$\text{for each } \nu \in \mathcal{E}, \quad \mathbb{P}_\nu[\tau < \infty \cap \hat{k} \notin \mathcal{A}_\epsilon(\nu)] \leqslant \delta, \tag{10}$$

*where* $\tau = \tau(\nu) \leqslant \infty$ *is the random stopping time of* $\pi$ *on instance* $\nu$, *arm* $\hat{k}$ *is the output upon termination, and* $\mathcal{A}_\epsilon(\nu)$ *is as defined in* (2).

**Remark 7** *This definition is related to the notion of being* $(\epsilon, \delta)$-*PAC (see (Even-Dar et al., 2002)). It can be seen as a relaxation of* $(\epsilon, \delta)$-*PAC since an* $(\epsilon, \delta)$-*reliable algorithm is allowed to be non-terminating on some instances – a high probability of correctness is needed only on instances it terminates on. As we will see in Section 5, this relaxation is required when considering every possible* $\nu \in \mathcal{E}$, *as there are instances that are not "solvable" for any finite number of arm pulls.*

**Theorem 8 (Reliability of Algorithm 1)** *Fix an instance* $\nu \in \mathcal{E}$, *and suppose Algorithm 1 is run with input* $(\mathcal{A}, \lambda, \epsilon, q, \delta)$ *and parameter* $c \geqslant 1$. *Under Event E as defined in Lemma 2, if Algorithm 1 terminates, then it returns an arm* $\hat{k}$ *satisfying* (2).

Since Event $E$ occurs with probability at least $1 - \delta$ (Lemma 2), we conclude the following.

**Corollary 9** *Algorithm 1 is* $(\epsilon, \delta)$-*reliable.*

The proof details of Theorem 8 are given in Appendix C, and we provide a sketch here. Combining the guarantee from Line 17, inequalities (7), and the choice of $\tilde{\epsilon} \leqslant \lambda \cdot \epsilon/((c+1)\lambda) = \epsilon/(c+1)$ yields

$$Q_{\hat{k}}(q) > \text{LCB}_t(\hat{k}) \geqslant \max_{a \in \mathcal{A}_t \setminus \{\hat{k}\}} \text{UCB}_t(a) - (c+1)\tilde{\epsilon} \geqslant \max_{a \in \mathcal{A}_t \setminus \{\hat{k}\}} Q_a(q) - \epsilon. \tag{11}$$

It remains to show that the optimal arm $k^*$ lies in $\mathcal{A}_t$ for all $t$, which we defer to Appendix C.

### 3.4. Upper Bound

In this section, we bound the number of arm pulls for a given instance $\nu \in \mathcal{E}$. To characterize the number of arm pulls, we define the gap of each arm as follows.

**Definition 10 (Arm gaps)** *Fix an instance* $\nu \in \mathcal{E}$. *Let* $\tilde{\epsilon} = \tilde{\epsilon}(\lambda, \epsilon, c)$ *and* $\mathcal{A}_\epsilon = \mathcal{A}_\epsilon(\nu)$ *be as defined in Algorithm 1 and* (2) *respectively. For each arm* $k \in \mathcal{A}$, *we define the gap* $\Delta_k = \Delta_k(\nu, \lambda, \epsilon, c, q)$ *as follows:*

$$\Delta_k := \begin{cases} \sup \left\{ \Delta \in [0, \min(q, 1-q)] : Q_k(q + \Delta) \leqslant \max_{a \in \mathcal{A}} Q_a^+(q - \Delta) - \tilde{\epsilon} \right\} & \text{if } k \notin \mathcal{A}_\epsilon \\ \max_{\mathcal{A}_\epsilon \subseteq S \subseteq \mathcal{A}} \Delta_k^{(S)} & \text{if } k \in \mathcal{A}_\epsilon \end{cases}, \tag{12}$$

*where* $Q_k^+(p)$ *is the upper quantile function defined in Lemma 4, and*

$$\Delta_k^{(S)} := \sup \left\{ \Delta \in \left[ 0, \min_{a \notin S} \Delta_a \right] : Q_k^+(q - \Delta) \geqslant \max_{a \in S \setminus \{k\}} Q_a(q + \Delta) - c\tilde{\epsilon} \right\} \tag{13}$$

*for each subset $S$ satisfying $\mathcal{A}_\epsilon \subseteq S \subseteq \mathcal{A}$. We use the convention that the minimum (resp. maximum) of an empty set is $\infty$ (resp. $-\infty$).*

**Remark 11 (Intuition on arm gaps)** *We provide some intuition for the gap definitions:*

- *For an arm $k \notin \mathcal{A}_\epsilon$, the gap $\Delta_k$ captures how much worse $k$ is than some other arm $a$. When $k$ is sufficiently pulled relative to $1/\Delta_k$, we can establish that $\mathrm{UCB}_t(k) \leqslant \mathrm{LCB}_t(a)$, which implies that $k$ is suboptimal, and we can stop pulling it. The details and derivation are given in Lemma 33 and its proof.*

- *To understand the gap $\Delta_k = \max_{\mathcal{A}_\epsilon \subseteq S \subseteq \mathcal{A}} \Delta_k^{(S)}$ for a satisfying arm $k \in \mathcal{A}_\epsilon$, we first consider $\Delta_k^{(S)}$ for a fixed subset $S \supseteq \mathcal{A}_\epsilon$. This captures how much better arm $k$ is than the "best" arm $a \in S$ (up to $\epsilon$). When arm $k$ is sufficiently pulled relative to $1/\Delta_k^{(S)}$, we can establish that the termination condition is satisfied. Since $S \supseteq \mathcal{A}_\epsilon$ is arbitrary, we define $\Delta_k$ based on the set $S$ giving the highest $\Delta_k^{(S)}$. The details and derivation are given in Lemma 35 and its proof.*

- *When some arm $k^*$ is the only satisfying arm (i.e., $\mathcal{A}_\epsilon = \{k^*\}$), we have*

$$\Delta_{k^*} \geqslant \Delta_{k^*}^{(\mathcal{A}_\epsilon)} = \sup\left\{\Delta \in \left[0, \min_{a \notin \mathcal{A}_\epsilon} \Delta_a\right] : Q_k^+(q-\Delta) \geqslant -\infty\right\} = \min_{a \notin \mathcal{A}_\epsilon} \Delta_a = \min_{a \neq k^*} \Delta_a. \quad (14)$$

  *This indicates that $k^*$ is pulled at most as many times as the smallest $\Delta_a$ value would dictate, and possibly fewer (if the while-loop terminates before $|\mathcal{A}_t| = 1$).*

**Remark 12 (Generalization and improvement over existing arm gap)** *Our gap definitions were developed with the view of ensuring that we can solve essentially all solvable instances, and we will establish results of this type in Section 5. Achieving this goal required several subtle choices in our gap definition, including the parameter $c$ and the optimization over $S$. We generalize existing gaps for the QMAB problem in the sense that those are recovered by considering $c \to \infty$, $S = \mathcal{A}$, and using only lower quantile functions. In Appendix D, we provide more details about these choices and give an instance where the gap is positive under our definition but is zero using existing definitions.*

**Remark 13 (Further improvement)** *Due to the assumption that the $q$-quantile of each arm is in $[0, \lambda]$, we can improve our gap definition by replacing the terms $Q_{(\cdot)}^+(q - \Delta)$ and $Q_{(\cdot)}(q + \Delta)$ with $\max\left\{0, Q_{(\cdot)}^+(q-\Delta)\right\}$ and $\min\left\{\lambda, Q_{(\cdot)}(q+\Delta)\right\}$ respectively. We adopt Definition 10 to avoid further complicating the gap definition and subsequent analysis, but we will provide detailed discussion of this modified gap in Appendix H.*

Having defined the arm gaps, we now state an upper bound on the total number of arm pulls by Algorithm 1.

**Theorem 14 (Upper bound)** *Fix an instance $\nu \in \mathcal{E}$, and suppose Algorithm 1 is run with input $(\mathcal{A}, \lambda, \epsilon, q, \delta)$ and parameter $c \geqslant 1$. Let $\mathcal{A}_\epsilon(\nu)$ be as defined in (2) and let the gap $\Delta_k = \Delta_k(\nu, \lambda, \epsilon, c, q)$*

*be as defined in Definition [10] for each arm $k \in \mathcal{A}$. Under Event $E$ as defined in Lemma [2], the total number of arm pulls is upper bounded by*

$$O\left(\left(\sum_{k \in \mathcal{A}} \frac{1}{\max\left(\Delta_k, \Delta\right)^2} \cdot \left(\log\left(\frac{1}{\delta}\right) + \log\left(\frac{1}{\max\left(\Delta_k, \Delta\right)}\right) + \log\left(\frac{c\lambda K}{\epsilon}\right)\right)\right)\right), \quad (15)$$

*where $\Delta = \Delta(\nu, \lambda, \epsilon, c, q) = \max_{a \in \mathcal{A}_\epsilon(\nu)} \Delta_a$.*

Combining Lemma [2], Theorem [8], and Theorem [14], we obtain the high-probability correctness for instances with positive gap.

**Corollary 15** *Fix an instance $\nu \in \mathcal{E}$ and suppose $\Delta = \Delta(\nu, \lambda, \epsilon, c, q)$ as defined in Theorem [14] is positive. Then, with probability at least $1 - \delta$, Algorithm [1] returns an arm $\hat{k}$ satisfying (2) and uses a total number of arm pulls satisfying (15).*

We will provide near-matching lower bounds in the next section, and an impossibility result for the instances with zero gap in Section [5]. The proof details of Theorem [14] are given in Appendix [E], and we provide a sketch here.

**Proof** [Proof outline for Theorem [14]] Under Event $E$, the while-loop of Algorithm [1] terminates when the round index $t$ is large enough to satisfy $\Delta^{(t)} \leqslant \frac{1}{2}\Delta$, which happens when $t = \log_2(1/\Delta) + \Theta(1)$ since $\Delta^{(t)} = 2^{-t+1}\min(q, 1-q)$. Summing through the number of arm pulls $\widetilde{O}\left(\left(\Delta^{(t)}\right)^{-2}\right)$ given in (6) for $t = 1, \ldots, \log_2(1/\Delta) + \Theta(1)$ yields the upper bound $\widetilde{O}\left(\Delta^{-2}\right)$ for each arm $k \in \mathcal{A}$. However, it is also possible that some arms are eliminated before the while-loop terminates. Specifically, each non-satisfying arm $k \notin \mathcal{A}_\epsilon(\nu)$ is eliminated when the index $t$ satisfies $\Delta^{(t)} \leqslant \frac{1}{2}\Delta_k$, which yields the upper bound $\widetilde{O}\left(\Delta_k^{-2}\right)$. Taking the minimum between these two gives (15). ■

## 4. Lower Bounds

In this section, we provide two lower bounds on the number of arm pulls. In Section [4.1], we provide a near-matching worst-case lower bound $\Omega\left(\sum_{k \in \mathcal{A}} \Delta_k^{-2}\log(\delta^{-1})\right)$ for instances with positive gap and $\epsilon$ is small enough such that $\mathcal{A}_\epsilon(\nu) = \{k^*\}$. This lower bounds holds even in the absence of communication constraints. In Section [4.2], we address the $\log(\lambda/\epsilon)$ dependence in the upper bound by showing that $\Omega(\log(\lambda/\epsilon))$ arm pulls are needed for any $(\epsilon, \delta)$-reliable algorithm when 1-bit threshold queries are used; in particular, targeting $\epsilon = 0$ is infeasible without further assumptions.

### 4.1. Lower Bound for the Unquantized Variant

We present a worst-case lower bound on the expected number of arm pulls for the setup with no communication constraint. The lower bound is based on a bad instance adapted from (Nikolakakis

et al., 2021, Theorem 4), which is for the quantile bandit problem of identifying the unique optimal arm $k^*$. Specifically, for instances with satisfying arm set $\mathcal{A}_\epsilon(\nu) = \{k^*\}$, the only correct output in both problem formulations is $k^*$. By choosing $\epsilon$ to be sufficiently small, the hard instance in their problem formulation (which does not allow an $\epsilon$ relaxation) can be adapted to be a hard instance in our problem formulation.

**Theorem 16 (Worst-case lower bound)** *Fix $q, \delta \in (0, 1)$ and $\lambda \geqslant 1$. There exists a quantile bandit instance $\nu \in \mathcal{E}$ with a unique best arm $k^*$ such that for any $\epsilon > 0$ satisfying*

$$\epsilon \leqslant \frac{1}{2} \big( Q_{k*}(q) - \max_{k \neq k*} Q_k(q) \big) \tag{16}$$

*and any $(\epsilon, \delta)$-reliable algorithm, the number of arm pulls $\tau$ satisfies*

$$\mathbb{E}[\tau] \geqslant \Omega \bigg( \sum_{k=1}^K \frac{1}{\Delta_k^2} \log \bigg( \frac{1}{\delta} \bigg) \bigg), \tag{17}$$

*where $\Delta_k(\nu, \epsilon, q) := \lim_{c \to \infty} \Delta_k(\nu, \lambda, \epsilon, c, q)$ is the gap defined in Definition 10 with $c \to \infty$ (see (43) for the explicit form).*

**Proof** See Appendix F.1. ∎

The only difference in the upper bound (15) and lower bound is that the lower bound only contains the log factor $\log(\delta^{-1})$ rather than the sum of three log factors, and so our upper bound matches the dependence on $\Delta_k$ of the lower bound to within a logarithmic factor. We note that if $\delta \leqslant \max(\Delta_k, \Delta)^{\Theta(1)}$ and $\delta \leqslant \big( \frac{\epsilon}{c\lambda K} \big)^{\Theta(1)}$ then the sum of three log terms in (15) simplifies to $O\big( \log(\delta^{-1}) \big)$, so in this "low error probability" regime we in fact get matching scaling laws in the upper and lower bound.

### 4.2. $\Omega(\log(\lambda/\epsilon))$ **Dependence Under Threshold Query Model**

In this section, we show that $\Omega(\log(\lambda/\epsilon))$ arm pulls is needed for any $(\epsilon, \delta)$-reliable algorithm in the case that only threshold queries are allowed. That is, the side information sent by the learner to the agent is always some threshold query of the form "Is $r_{a_t,t} \leqslant \gamma_t$?", and the learner receives the 1-bit comparison feedback $\mathbf{1}(r_{a_t,t} \leqslant \gamma_t)$. This is a common 1-bit quantization method in practice and is also the one used in Algorithm 1, though it would also be of interest to determine whether using other 1-bit quantization methods can help.

**Theorem 17 ($\Omega(\log(\lambda/\epsilon))$ dependence)** *Fix $\lambda \geqslant \epsilon > 0$, and $q \in (0, 1)$, and $\delta \in (0, 0.5)$. Under the threshold query model, there exists a two-arm quantile bandit instance $\nu$ with deterministic rewards such that any $(\epsilon, \delta)$-reliable algorithm requires $\Omega(\log(\lambda/\epsilon))$ arm pulls.*

**Proof** The idea is that if the two deterministic arms in $[0, \lambda]$ are separated by $2\epsilon$, then a binary search over $\Theta(\lambda/\epsilon)$ possible choices is needed just to locate them. See Appendix F.2 for the details. ∎

While our upper and lower bounds match to within at most logarithmic factors under mild conditions, we leave it open as to (i) whether the dependence on $\Delta_k$ can be improved in general (e.g., to doubly-logarithmic as in the ones in unquantized quantile BAI (Nikolakakis et al., 2021; Howard and Ramdas, 2022), and (ii) whether there exist regimes in which the *joint* dependence on the gaps and $(\lambda, \epsilon)$ can be improved.

## 5. Solvable Instances

In Sections 3.4 and 4.1, we provided nearly matching upper and lower bounds for instances with positive gap. In this section, we study the "(un)solvability" of bandit instances with zero gap, and show that essentially all bandit instances that are "solvable" have positive gap, as long as parameter $c$ is large enough (see Remark 23). To formalize this idea, we define the following class of bandit instances.

**Definition 18 (Solvable instances)** *Let $\mathcal{A}, \epsilon$, and $q$ be fixed. We say that an instance $\nu \in \mathcal{E}$ is $\epsilon$-solvable if for each $\delta \in (0, 1)$, there exists an algorithm that is $(\epsilon, \delta)$-reliable and it holds under instance $\nu$ that*[8]

$$\mathbb{P}_\nu[\tau < \infty \cap \hat{k} \in \mathcal{A}_\epsilon] \geqslant 1 - \delta. \tag{18}$$

*If no such algorithm exists, we say that $\nu$ is $\epsilon$-unsolvable.*

**Remark 19** *Fix $0 < \epsilon_1 \leqslant \epsilon_2$. If an instance $\nu$ is $\epsilon_1$-solvable, then it is $\epsilon_2$-solvable. This follows directly from $\mathcal{A}_{\epsilon_1}(\nu) \subseteq \mathcal{A}_{\epsilon_2}(\nu)$.*

From Corollary 15, we deduce that any instance with a positive gap is solvable.

**Corollary 20 (Positive gap is solvable)** *Let $\mathcal{A}, \lambda, \epsilon, q$, and $c$ be fixed. Suppose an instance $\nu$ satisfies $\Delta > 0$, where $\Delta = \Delta(\nu, \lambda, \epsilon, c, q)$ is as defined in Theorem 14. Then $\nu$ is $\epsilon$-solvable.*

The main result of this section is that the reverse inclusion nearly holds, in the following sense.

**Theorem 21 (Zero gap is unsolvable)** *Let $\lambda, \epsilon, c$, and $q$ be fixed, and let $\tilde{\epsilon} = \tilde{\epsilon}(\lambda, \epsilon, c)$ be as defined in Algorithm 1. Suppose an instance $\nu \in \mathcal{E}$ satisfies $\Delta(\nu, \lambda, \epsilon, c, q) = 0$. If we assume for $\nu$ that there exists some sufficiently small $\eta_0 > 0$ such that $0 \leqslant Q_k^+(q - \eta_0) \leqslant Q_k(q + \eta_0) \leqslant \lambda$, then $\nu$ is $c\tilde{\epsilon}$-unsolvable.*

---

8. We could require that $\mathbb{P}_\nu[\tau < \infty] = 1$ in this case and the subsequent analysis and conclusions would be essentially unchanged. Recall also that $\mathbb{P}_\nu[\cdot]$ denotes probability under instance $\nu$.

**Proof** See Appendix G. ∎

**Remark 22 (Removing the additional assumption)** *The additional assumption involving $\eta_0$ is mild; it is trivially satisfied by instances with all reward distributions supported on $[0, \lambda]$, and also holds significantly more generally since $\eta_0$ can be arbitrarily small. Moreover, in Appendix H.3, we show that this assumption is unnecessary if we use the modified gap (see Remark 13) instead of $\Delta$.*

**Remark 23** *For each $\theta \in (0, 1)$, picking $c = \lceil 2\theta/(1 - \theta) \rceil$ yields*

$$\nu \text{ is } \theta\epsilon\text{-solvable} \implies \nu \text{ is } c\tilde{\epsilon}\text{-solvable} \implies \Delta(\nu, \lambda, \epsilon, c, q) > 0 \implies \nu \text{ is } \epsilon\text{-solvable}, \quad (19)$$

*where the last two implications follow from Theorem 21 and Corollary 20, and the first implication follows from Remark 19 and the following inequality:*

$$c\tilde{\epsilon} = \frac{c\lambda}{\lceil (c+1)\lambda/\epsilon \rceil} \geqslant \frac{c\lambda}{(c+2)\lambda/\epsilon} = \left(1 - \frac{2}{c+2}\right)\epsilon \geqslant \left(1 - \frac{2}{\frac{2\theta+2-2\theta}{1-\theta}}\right)\epsilon = \theta\epsilon. \quad (20)$$

*Since $\theta$ can be arbitrarily close to 1, we have $\Delta(\nu, \lambda, \epsilon, c, q) > 0$ for essentially all $\epsilon$-solvable instances by picking a sufficiently large $c$.*

The proof of Theorem 21 will turn out to directly extend to a "limiting" version in which we replace $c\tilde{\epsilon}$ by $\lim_{c \to \infty} c\tilde{\epsilon} = \epsilon$ and $\Delta(\nu, \lambda, \epsilon, c, q)$ by $\lim_{c \to \infty} \Delta(\nu, \lambda, \epsilon, c, q)$, giving the following corollary.

**Corollary 24** *Let $\lambda$, $\epsilon$, and $q$ be fixed. Let $\Delta_k(\nu, \epsilon, q)$ be the gap defined in Definition 10 with $c \to \infty$ (see (43) for the explicit form). Suppose an instance $\nu \in \mathcal{E}$ satisfies $\Delta(\nu, \epsilon, q) = \max_{k \in \mathcal{A}_{\epsilon(\nu)}} \Delta_k(\nu, \epsilon, q) = 0$. If we assume for $\nu$ that there exists some sufficiently small $\eta_0 > 0$ such that $0 \leqslant Q_k^+(q - \eta_0) \leqslant Q_k(q + \eta_0) \leqslant \lambda$, then $\nu$ is $\epsilon$-unsolvable.*

**Proof** See Appendix G. ∎

## Acknowledgments

This work was supported by the Singapore National Research Foundation under its AI Visiting Professorship programme.

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

## Appendix A. Quantile Estimation Subroutine

### A.1. Noisy Binary Search

We first momentarily depart from MAB and discuss the monotonic noisy binary search (MNBS) problem of (Karp and Kleinberg, 2007); see also the end of Appendix A.2 for a summary of some related work on noisy binary search. The original problem formulation was stated in terms of finding a special coin $i$ among $n$ coins, but this can be restated as follows: We have a random variable $R$ with an unknown CDF $F$ and a list of $n$ points $x_1 \leqslant \cdots \leqslant x_n$ such that $Q(\tau) \in [x_1, x_n]$, and the goal is to find an index $i$ satisfying

$$[F(x_i), F(x_{i+1})] \cap (\tau - \Delta, \tau + \Delta) \neq \varnothing \tag{21}$$

via adaptive queries of the form $\mathbf{1}(R \leqslant x_j)$. Note that each query $\mathbf{1}(R \leqslant x_j)$ is an independent Bernoulli random variable with parameter $F(x_j)$. We will make use of the following main result from (Gretta and Price, 2024, Theorem 1.1).

**Proposition 25 (Noisy binary search guarantee)** *For any $\delta \in (0,1)$ and relaxation parameter $\Delta \leqslant \min(\tau, 1 - \tau)$, the MNBS algorithm in (Gretta and Price, 2024) output an index $i$ after at most $O\left(\frac{1}{\Delta^2} \log \frac{n}{\delta}\right)$ queries[9] and $i$ satisfies (21) with probability at least $1 - \delta$.*

The bulk of the MNBS algorithm in (Gretta and Price, 2024) is based on Bayesian multiplicative weight updates: Start with a uniform prior over which of the $n$ intervals crosses quantile $\tau$, make the query at $x_j$ whose $F(x_j)$ is nearest to $\tau$ under current distribution, update the posterior by multiplying intervals on one side of the query by $1 + c\Delta$ and the other side by $1 - c\Delta$ for some fixed constant $c$, and repeat. Other MNBS algorithms such as those in (Karp and Kleinberg, 2007), or even a naive binary search with repetitions (see (Karp and Kleinberg, 2007, §1.2)), could also be used to solve the MNBS problem, but we choose (Gretta and Price, 2024) since it has the best known scaling of the query complexity. Further comparisons of the relevant theoretical guarantees and practical performance can be found in (Gretta and Price, 2024).

### A.2. Quantile Estimation with 1-bit Feedback

The MNBS algorithm can be implemented under our 1-bit communication-constrained setup. Specifically, the learner decides which arm $k$ to query as well as the point $x_j$ to query, and then sends a threshold query "Is $R_k \leqslant x_j$?" as side information to the agent, where $R_k$ is the random reward (variable) of the arm $k$ with CDF $F_k$. The agent will then pull arm $k$ and reply with a 1-bit binary feedback corresponding to the observation. Note that while the $O\left(\frac{1}{\Delta^2} \log \frac{n}{\delta}\right)$ queries for a given arm are done in an adaptive manner, the queries themselves can be requested at different time steps without any requirement of agent memory. A high-level description of the implementation for a fixed arm is given in Algorithm 2. This gives us the following guarantee, which is a simple consequence of Proposition 25.

---

9. The expression for the number of iterations in (Gretta and Price, 2024) is more complicated because it has some terms with explicit constant factors, but in $O(\cdot)$ notation it simplifies to $O\left(\frac{1}{\Delta^2} \log \frac{n}{\delta}\right)$. We do not specify the exact number of loops in Algorithm 2, as doing so is somewhat cumbersome and the focus of our work is on the scaling laws.

---

**Algorithm 2** Communication-constrained quantile estimation subroutine (`QuantEst` in Algorithm 1)

---

**Input**: Arm with reward $R$ distributed according to CDF $F$, a list $\mathbf{x}$ of $n$ points $x_1 \leqslant \cdots \leqslant x_n$, quantile $\tau \in (0,1)$ satisfying $Q(\tau) \in [x_1, x_n]$, approximation parameter $\Delta \leqslant \min(\tau, 1-\tau)$, error probability $\delta \in (0,1)$

**Output** Index $i \in \{1, \ldots, n-1\}$

1: **for** $t = 1$ to $t_{\max}$ (with[9] $t_{\max} = O\left(\frac{1}{\Delta^2} \log \frac{n}{\delta}\right)$) **do**
2:     **At Learner:**
3:         Pick index $j$ according to Bayesian weight update as in (Gretta and Price, 2024)
4:         Send threshold query "Is $R \leqslant x_j$?" to the agent
5:     **At Agent:**
6:         Pull arm and observe reward $r$
7:         Send 1-bit feedback $\mathbf{1}(r \leqslant x_j)$ to the learner
8: Return index $i$ according to (Gretta and Price, 2024)

---

**Corollary 26** (`QuantEst` **guarantee**) *Let $(F, \mathbf{x}, \tau, \Delta, \delta)$ be a valid input of Algorithm 2, and let $n$ be the number of points in $\mathbf{x}$. Then the algorithm outputs an index $i$ after at most $O\left(\frac{1}{\Delta^2} \log \frac{n}{\delta}\right)$ queries and $i$ satisfies $\mathbb{P}\left([F(x_i), F(x_{i+1})] \cap (\tau - \Delta, \tau + \Delta) = \varnothing\right) < \delta$.*

**Related work on noisy binary search and quantile estimation.** We briefly recap the original MNBS problem in (Karp and Kleinberg, 2007; Gretta and Price, 2024): There are $n$ coins whose unknown probabilities $p_j \in [0, 1]$ are sorted in nondecreasing order, where flipping coin $j$ results in head with probability $p_j$. The goal is to identify a coin $i$ such that the interval $[p_i, p_{i+1}]$ has a nonempty intersection with $(\tau - \Delta, \tau + \Delta)$. This model subsumes noisy binary search with a fixed noise level (Burnashev and Zigangirov, 1974; Ben-Or and Hassidim, 2008; Dereniowski et al., 2021; Gu and Xu, 2023) (where $p_j = \frac{1}{2} - \Delta$ for $j \leqslant i$ and $p_j = \frac{1}{2} + \Delta$ otherwise) as well as regular binary search (where $p_j \in \{0, 1\}$). As we discussed in Appendix A.1, this problem can be reformulated into the problem of estimating (the quantile of) a distribution using threshold/comparison queries, where the noise in the feedback is stochastic. This quantile estimation problem has been generalized to a non-stochastic noise setting (Meister and Nietert, 2021; Okoroafor et al., 2023), and was also studied in the context of online dynamic pricing and auctions (Kleinberg and Leighton, 2003; Paes Leme et al., 2023b,a). In particular, (Paes Leme et al., 2023b, Algorithm 1) is similar to Algorithm 2 (or equivalently subroutine `QuantEst` used on Lines 11 and 13 of Algorithm 1), in the sense that both use noisy binary search to identify the quantile of a *single* distribution. However, they use the naive binary search with repetitions to form confidence intervals containing the quantile, which has a suboptimal complexity $O\left(\frac{1}{\Delta^2} \log n \log \frac{\log n}{\delta}\right)$; see (Karp and Kleinberg, 2007, §1.2) for details. Overall, while ideas from the existing literature on quantile estimation of a *single* distribution with threshold queries may provide useful context, they do not readily translate into Algorithm 1 or the analysis that led to our main contributions.

### A.3. Proof of Lemma 2 (Bounding the Probability of Event E)

**Proof** [Proof of Theorem 2] For a fixed $t \geqslant 1$ and a fixed $k \in \mathcal{A}_t$, we have

$$\mathbb{P}\left(\overline{E_{t,k,l}}\right) \leqslant \frac{\delta \cdot \Delta^{(t)}}{2|\mathcal{A}_t|} \quad \text{and} \quad \mathbb{P}\left(\overline{E_{t,k,u}}\right) \leqslant \frac{\delta \cdot \Delta^{(t)}}{2|\mathcal{A}_t|} \tag{22}$$

by the guarantee of the QuantEst (see Corollary 26). Applying the union bound, we obtain

$$\mathbb{P}\left(\overline{E}\right) \leqslant \sum_{t \geqslant 1} \sum_{k \in \mathcal{A}_t} \frac{\delta \cdot \Delta^{(t)}}{|\mathcal{A}_t|} \leqslant \sum_{t \geqslant 1} |\mathcal{A}_t| \cdot \frac{\delta \cdot \Delta^{(t)}}{|\mathcal{A}_t|} = \delta \sum_{t \geqslant 1} \Delta^{(t)} = \delta \sum_{t \geqslant 1} 2^{-t} \leqslant \delta \tag{23}$$

as desired. The number of arm pulls (6) follows immediately from the guarantee of QuantEst from Corollary 26, $|\mathcal{A}_t| \leqslant |\mathcal{A}| = K$, and the number of points $n = \Theta(c\lambda/\epsilon)$. ∎

## Appendix B. Proof of Lemma 4 (Anytime Quantile Bounds)

We first present a useful auxiliary lemma.

**Lemma 27** *Under the setup of Lemma 4 (including Event E from Lemma 2 holding), we have the following bounds:*

$$x_{l_{t,k}} < Q_k(q) \tag{24}$$

$$Q_k^+\left(q - \Delta^{(t)}\right) \leqslant x_{l_{t,k}+1} \tag{25}$$

$$x_{u_{t,k}} < Q_k\left(q + \Delta^{(t)}\right) \tag{26}$$

$$Q_k^+(q) \leqslant x_{u_{t,k}+1} \tag{27}$$

*for each round $t \geqslant 1$ and arm $k \in \mathcal{A}_t$.*

**Proof** We will prove only (24) and (25) for an arbitrary $t \geqslant 1$ and $k \in \mathcal{A}_t$ in detail, as (26) and (27) can be proved similarly. Observe that, under event $E_{t,k,l} \subset E$ (see (3)), we have

$$F_k(x_{l_{t,k}}) < q \quad \text{and} \quad q - \Delta^{(t)} < F_k(x_{l_{t,k}+1}) \tag{28}$$

respectively, as otherwise the interval $\left[F_k(x_{l_{t,k}}), F_k(x_{l_{t,k}+1})\right]$ would fall on the right and the left, respectively, of the interval $\left(q - \Delta^{(t)}, q\right)$. A similar argument through the event $E_{t,k,u} \subset E$ (see (4)) yields

$$F_k(x_{u_{t,k}}) < q + \Delta^{(t)} \quad \text{and} \quad q < F_k(x_{u_{t,k}+1}). \tag{29}$$

We now prove (24) using (28); the inequality (26) can be proved similarly through (29). If $x_{l_{t,k}} = -\infty$, then (24) holds trivially. Therefore, we proceed on the assumption that $x_{l_{t,k}} \in \mathbb{R}$. Then, using standard properties of quantile functions (see, e.g., (Dufour, 1995, 4.3 Theorem)), we have $x_{l_{t,k}} < Q_k(q)$ as desired.

We now prove (25) using (28); the inequality (27) can be proved similarly through (29). If $x_{l_{t,k}+1} = \infty$, then (25) holds trivially. Therefore, we proceed on the assumption that $x_{l_{t,k}+1} \in \mathbb{R}$. In this case, it is a finite upper bound on the values in the set $\{z \in \mathbb{R} : F_k(z) \leqslant q - \Delta^{(t)}\}$, and so this set has a finite supremum. It follows that

$$x_{l_{t,k}+1} \geqslant \sup\{z \in \mathbb{R} : F_k(z) \leqslant q - \Delta^{(t)}\} = Q_k^+\big(q - \Delta^{(t)}\big) \tag{30}$$

as desired. ■

**Proof** [Proof of Lemma 4] We break down the bounds into inequalities as follows:

(i) $\mathrm{LCB}_\tau(k) \leqslant \mathrm{LCB}_t(k)$     (iv) $\mathrm{UCB}_t(k) \leqslant \mathrm{UCB}_\tau(k)$

(ii) $\mathrm{LCB}_t(k) < Q_k(q)$      (v) $Q_k^+\big(q - \Delta^{(t)}\big) \leqslant \mathrm{LCB}_t(k) + \tilde{\epsilon}$

(iii) $Q_k(q) \leqslant \mathrm{UCB}_t(k)$      (vi) $\mathrm{UCB}_t(k) - \tilde{\epsilon} < Q_k\big(q + \Delta^{(t)}\big)$

We will prove only inequalities (i), (ii), and (iv) for an arbitrary $t > \tau \geqslant 0$ and $k \in \mathcal{A}_t$ in detail, as all the other inequalities can be proved similarly.

Inequality (i) follows immediately from Line 12 of Algorithm 1 and induction. Likewise, we can show (iv) using Line 14 of Algorithm 1.

We now show inequality (ii) by induction on $t$; inequality (iii) can be proved similarly. For the base case $t = 1$, we have

$$\mathrm{LCB}_1(k) = \max\big(x_{l_{t,k}}, \mathrm{LCB}_0(k)\big) = \max\big(x_{l_{t,k}}, 0\big) = x_{l_{t,k}} < Q_k(q), \tag{31}$$

where the last inequality follows from (24). For the inductive step, suppose that $\mathrm{LCB}_t(k) < Q_k(q)$ for a fixed $t \geqslant 1$. Since $x_{l_{t,k}} < Q_k(q)$, we have

$$\mathrm{LCB}_{t+1}(k) = \max\big(x_{l_{t,k}}, \mathrm{LCB}_t(k)\big) < Q_k(q) \tag{32}$$

as desired.

We now show inequality (v) using (25); inequality (vi) can be shown using a similar argument through (26). We consider three cases for the index $l_{t,k}$:

- $(l_{t,k} = 0)$ In this case, we have $x_{l_{t,k}+1} = x_1 = 0 = \mathrm{LCB}_0(k)$, and so

$$Q_k^+\big(q - \Delta^{(t)}\big) \leqslant x_{l_{t,k}+1} = \mathrm{LCB}_0(k) \leqslant \mathrm{LCB}_t(k) < \mathrm{LCB}_t(k) + \tilde{\epsilon}, \tag{33}$$

  where the first inequality follows from (25) and the second inequality follows from inequality (i).

- $(1 \leqslant l_{t,k} \leqslant n)$ In this case, we have

$$Q_k^+\big(q - \Delta^{(t)}\big) \leqslant x_{l_{t,k}+1} = x_{l_{t,k}} + \tilde{\epsilon} \leqslant \mathrm{LCB}_t(k) + \tilde{\epsilon}, \tag{34}$$

  where the first inequality follows from (25), the equality follows from distance between consecutive points set in Line 3 of Algorithm 1, and the last inequality follows from Line 12 of Algorithm 1.

- $(l_{t,k} = n + 1)$ In this case, we have $x_{l_{t,k}} = x_{n+1} = \lambda \geqslant Q_k(q) \geqslant Q_k^+(q - \Delta^{(t)})$, and so

$$Q_k^+(q - \Delta^{(t)}) \leqslant x_{l_{t,k}} \leqslant \text{LCB}_t(k) < \text{LCB}_t(k) + \tilde{\epsilon}, \tag{35}$$

where the second inequality follows from Line 12 of Algorithm 1.

Combining all three cases, we have $Q_k^+(q - \Delta^{(t)}) \leqslant \text{LCB}_t(k) + \tilde{\epsilon}$ as desired. ∎

## Appendix C. Proof of Theorem 8 (Reliability of Algorithm 1)

**Proof** [Proof of Theorem 8] We first show by induction that an optimal arm $k^*$ of instance $\nu$ (i.e., one having the highest $q$-quantile) will always be active, i.e., $k^* \in \mathcal{A}_t$ for each round $t \geqslant 1$. For the base case $t = 1$, we have $k^* \in \{1, \ldots, K\} = \mathcal{A}_1$ trivially. We now show the inductive step: if $k^* \in \mathcal{A}_t$ holds, then $k^* \in \mathcal{A}_{t+1}$. For all arms $a \in \mathcal{A}_t$, we have

$$\text{UCB}_t(k^*) \geqslant Q_{k*}(q) \geqslant Q_a(q) > \text{LCB}_t(a), \tag{36}$$

where the second inequality follows from the optimality of arm $k^*$, while the other two inequalities follow from the anytime quantile bounds (Lemma 4). It follows that $\text{UCB}_t(k^*) > \max_{a \in \mathcal{A}_t} \text{LCB}_t(a)$, and so $k^* \in \mathcal{A}_{t+1}$ by definition (see Line 15 of Algorithm 1).

We now argue that if Algorithm 1 terminates, then the returned arm $\hat{k}$ satisfies (2). If Algorithm 1 terminates, then the while-loop (Lines 8–16) must have terminated and therefore the returned arm $\hat{k}$ satisfies the condition

$$\text{LCB}_t(\hat{k}) \geqslant \max_{a \in \mathcal{A}_t \setminus \{\hat{k}\}} \text{UCB}_t(a) - (c + 1)\tilde{\epsilon} \geqslant \max_{a \in \mathcal{A}_t \setminus \{\hat{k}\}} \text{UCB}_t(a) - \epsilon, \tag{37}$$

where the second inequality follows from Lines 1–2 of Algorithm 1: $\tilde{\epsilon} \leqslant \lambda \cdot \epsilon/((c+1)\lambda) = \epsilon/(c+1)$. If $\hat{k} = k^*$, then the returned arm satisfies (2) trivially. Therefore, we assume that $\hat{k} \neq k^*$ for the rest of the proof. In this case, we have

$$Q_{\hat{k}}(q) > \text{LCB}_t(\hat{k}) \geqslant \max_{a \in \mathcal{A}_t \setminus \{\hat{k}\}} \text{UCB}_t(a) - \epsilon \geqslant \text{UCB}_t(k^*) - \epsilon \geqslant \max_{a \in \mathcal{A}_t \setminus \{\hat{k}\}} Q_a(q) - \epsilon. \tag{38}$$

where the first and the last inequalities follow from the anytime quantile bounds (see Lemma 4), while the second inequality follows from the condition (37) and the third inequality follows from $k^* \in \mathcal{A}_t$ (see above) and the assumption that $\hat{k} \neq k^*$. ∎

## Appendix D. Details on Remark 12 (Comparison to Existing Gap Definitions)

We first recall some existing arm gap definitions for the exact quantile bandit problem (i.e., $\epsilon = 0$) in the setting of unquantized rewards. In (Nikolakakis et al., 2021, Definition 2), the authors defined the gap $\Delta_k^{\text{NKSS}}$ for each suboptimal arm $k \neq k^*$ by

$$\Delta_k^{\text{NKSS}} := \sup\{\Delta \in [0, \min(q, 1 - q)] : Q_k(q + \Delta) \leqslant Q_{k*}(q - \Delta)\}. \tag{39}$$

While the authors did not define the arm gap for $k^*$, we can take it to be the same as the gap of the "best" suboptimal arm, as their algorithm terminates only when all suboptimal arms are eliminated. On the other hand, the arm gap defined in (Howard and Ramdas, 2022, (Eq. (27))) is given by

$$\Delta_k^{\mathrm{HR}} := \begin{cases} \sup\{\Delta \in [0, \min(q, 1-q)] : Q_k(q+\Delta) \leqslant \max_{a \in \mathcal{A}} Q_a(q-\Delta)\} & \text{if } k \neq k^* \\ \sup\{\Delta \in [0, q] : Q_k(q-\Delta) \geqslant \max_{a \neq k} Q_a(q + \Delta_a^{\mathrm{HR}})\} & \text{if } k = k^* \end{cases}. \quad (40)$$

Similar to our arm gap definition (Definition 10), the gaps $\Delta_k^{\mathrm{HR}}$ for suboptimal arms $k \neq k^*$ are not defined based on the quantile function of $k^*$. It follows that $\Delta_a^{\mathrm{HR}} \geqslant \Delta_a^{\mathrm{NKSS}}$ for all arms $a \in \mathcal{A}$.

We now study the effect of taking $c \to \infty$ in our gap, which is given below in (43). From (43), it is straightforward to verify that (40) is recovered from our gap (Definition 10) by using only lower quantile functions and taking $S = \mathcal{A}$ and $c \to \infty$.

**Effect of parameter $c$ in the gap definition.** For any $1 \leqslant c_1 \leqslant c_2$, let

$$\tilde{\epsilon}_1 = \frac{\lambda}{\lceil (c_1+1)\lambda/\epsilon \rceil} \quad \text{and} \quad \tilde{\epsilon}_2 = \frac{\lambda}{\lceil (c_2+1)\lambda/\epsilon \rceil} \quad (41)$$

be as defined using Lines 1–2 of in Algorithm 1. It can readily be verified that

$$\tilde{\epsilon}_1 \geqslant \tilde{\epsilon}_2 \quad \text{and} \quad c_1\tilde{\epsilon}_1 \leqslant c_2\tilde{\epsilon}_2 \leqslant \epsilon \quad \text{and} \quad \Delta_k(\nu, \lambda, \epsilon, c_1, q) \leqslant \Delta_k(\nu, \lambda, \epsilon, c_2, q). \quad (42)$$

Since $\lim_{c \to \infty} \tilde{\epsilon} = 0$ and $\lim_{c \to \infty} c\tilde{\epsilon} = \epsilon$, the gap as defined in Definition 10 converges to a quantity $\Delta_k := \Delta_k(\nu, \epsilon, q) = \lim_{c \to \infty} \Delta_k(\nu, \lambda, \epsilon, c, q)$, given by

$$\Delta_k = \begin{cases} \sup\left\{ \Delta \in [0, \min(q, 1-q)] : Q_k(q+\Delta) \leqslant \max_{a \in \mathcal{A}} Q_a^+(q-\Delta) \right\} & \text{if } k \notin \mathcal{A}_\epsilon \\ \max_{\mathcal{A}_\epsilon \subseteq S} \left\{ \sup\left\{ \Delta \in \left[0, \min_{a \notin S} \Delta_a\right] : Q_k^+(q-\Delta) \geqslant \max_{a \in S \setminus \{k\}} Q_a(q+\Delta) - \epsilon \right\} \right\} & \text{if } k \in \mathcal{A}_\epsilon \end{cases}. \quad (43)$$

Note that $\Delta_k$ is independent of $c$ and $\lambda$.

**Remark 28 (Use of upper quantile function)** *To our knowledge, we are the first to incorporate upper quantile functions in the gap definition. This may lead to a potentially larger arm gap as compared to defining using only lower quantile functions (e.g., changing $Q_a^+(\cdot)$ and $Q_k^+(\cdot)$ in (12) and (13) to $Q_a(\cdot)$ and $Q_k(\cdot)$ respectively), and hence a tighter upper bound.*

**Remark 29 (Dependency on $Q_{k*}(q-\Delta)$)** *Existing papers using an elimination-based algorithm have their arm gaps defined according to $Q_{k*}(q-\Delta)$; see (39) for an example. In contrast, we remove this dependency and define using $\max_{a \in \mathcal{A}} Q_a^+(q-\Delta)$, which may lead to a tighter upper bound. The resulting analysis required is more challenging – see the discussion in Remark 34.*

Since our gap definitions generalizes existing gap definitions, we expect that their gaps being positive on an instance $\nu$ would imply our gap being positive on $\nu$. That is, their gaps being positive is a sufficient condition for Algorithm 1 to return a satisfying arm with high-probability (see Corollary 15).

**Proposition 30** *Fix an instance $\nu \in \mathcal{E}$ that has a unique arm $k^*$ with the highest $q$-quantile. Let $\Delta_a^{\mathrm{NKSS}}$ and $\Delta_a^{\mathrm{HR}}$ be as defined in (39) and (40) for each $a \in \mathcal{A}$. If $\min_{a \in \mathcal{A}} \{\Delta_a^{\mathrm{NKSS}}\} > 0$ or $\min_{a \in \mathcal{A}} \{\Delta_a^{\mathrm{HR}}\} > 0$, then $\Delta = \Delta(\nu, \lambda, \epsilon, c, q)$ as defined in Theorem 14 is also positive.*

**Proof** It suffices to consider the case $\min_{a \in \mathcal{A}} \{\Delta_a^{\mathrm{HR}}\} > 0$, since $\Delta_a^{\mathrm{HR}} \geqslant \Delta_a^{\mathrm{NKSS}}$ for all arms $a \in \mathcal{A}$. Let $\eta = \frac{1}{2} \min_{a \in \mathcal{A}} \Delta_a^{\mathrm{HR}} > 0$. Then we have

$$Q_{k^*}^+(q - \eta) \geqslant Q_{k^*}(q - \eta) \geqslant \max_{a \neq k} Q_a(q + \Delta_a^{\mathrm{HR}}) \geqslant \max_{a \in \mathcal{A} \setminus \{k^*\}} Q_a(q + \eta) - c\tilde{\epsilon}, \qquad (44)$$

where the second inequality follows from (40) and $\tilde{\epsilon} = \tilde{\epsilon}(\lambda, \epsilon, c)$ is as defined in Algorithm 1. Combining (44) and (13) of our gap definition, we have

$$\max_{a \in \mathcal{A}_\epsilon} \Delta_a \geqslant \Delta_{k^*} = \max_{\mathcal{A}_\epsilon \subseteq S \subseteq \mathcal{A}} \Delta_{k^*}^{(S)} \geqslant \Delta_{k^*}^{(\mathcal{A})} \geqslant \eta > 0 \qquad (45)$$

as desired. ∎

We now show that the converse is not true in general. In other words, there exists an instance $\nu \in \mathcal{E}$ where no algorithm can distinguish which arm has a higher quantile using a finite number of arm pulls (see (Nikolakakis et al., 2021, Theorem 2)), but Algorithm 1 is capable of returning an $\epsilon$-satisfying arm with high probability.

**Proposition 31** *Fix $\lambda \geqslant \epsilon > 0$ and $\delta \in (0, 0.5)$. There exists a two-arm bandit instance $\nu \in \mathcal{E}$ that has a unique arm $k^*$ with the highest median such that $\Delta = \Delta(\nu, \lambda, \epsilon, c, q)$ as defined in Theorem 14 is positive for $c \geqslant 2$, but $\min_{a \in \mathcal{A}} \{\Delta_a^{\mathrm{NKSS}}\} = \min_{a \in \mathcal{A}} \{\Delta_a^{\mathrm{HR}}\} = 0$.*

**Proof** Consider two arms $\mathcal{A} = \{1, 2\}$ with the following CDFs:

$$F_1(x) = \begin{cases} 0 & \text{for } x < 0 \\ \frac{x}{2m_1} & \text{for } 0 \leqslant x < 2m_1 \\ 1 & \text{for } x \geqslant 2m_1 \end{cases} \quad \text{and} \quad F_2(x) = \begin{cases} 0 & \text{for } x < m_2 \\ 0.5 & \text{for } m_2 \leqslant x < 2m_1 \\ 1 & \text{for } x \geqslant 2m_1 \end{cases}, \qquad (46)$$

where $m_2 \in (m_1 - \epsilon/2, m_1)$ such that both arms are $\epsilon$-optimal, with arm 1 being the unique best arm. Note that for each $\eta > 0$, we have

$$Q_2(0.5 + \eta) = 2m_1 > m_1 = Q_1(0.5) \geqslant Q_1(0.5 - \eta), \qquad (47)$$

and so $\Delta_2^{\mathrm{NKSS}} = \Delta_2^{\mathrm{HR}} = 0$. However, under our gap definition (Definition 10) with $\mathcal{A}_\epsilon(\nu) = \{1, 2\} = \mathcal{A}$ and any $c \geqslant 2$, we have

$$\Delta \geqslant \Delta_2 \geqslant \Delta_2^{(\{1,2\})} = \sup \left\{ \Delta \in [0, 0.5] : Q_2^+(0.5 - \Delta) \geqslant Q_1(0.5 + \Delta) - c\tilde{\epsilon} \right\} \qquad (48)$$

$$\geqslant \sup \left\{ \Delta \in [0, 0.5] : Q_2^+(0.5 - \Delta) \geqslant Q_1(0.5 + \Delta) - \frac{\epsilon}{2} \right\} \qquad (49)$$

$$= \sup \left\{ \Delta \in [0, 0.5] : m_2 \geqslant (1 + 2\Delta)m_1 - \frac{\epsilon}{2} \right\} \qquad (50)$$

$$= \min \left\{ 0.5, \frac{m_2 - (m_1 - \epsilon/2)}{2m_1} \right\} > 0, \qquad (51)$$

where the second inequality follows from the calculation in Remark 23, and the last inequality follows from the assumptions that $m_1 > 0$ and $m_2 > m_1 - \epsilon/2$. ∎

## Appendix E. Proof of Theorem 14 (Upper Bound of Algorithm 1)

We break down the upper bound on the number of arm pulls used by Algorithm 1 as follows. We bound the number of rounds required for a non-satisfying arm $k \notin \mathcal{A}_\epsilon(\nu)$ to be eliminated in Lemma 33. Then in Lemma 35, we bound the number of rounds each non-eliminated arm has gone through when the termination condition of the while-loop is triggered. Combining these lemmas with the number of arm pulls used by QuantEst for each round index $t \geqslant 1$ and active arm $k \in \mathcal{A}_t$ as stated in (6) gives us an upper bound on the total number of arm pulls.

We first present a useful lemma that will be used in the proofs of the two subsequent lemmas.

**Lemma 32 ($\max \mathrm{LCB}$ is non-decreasing)** *Under Event $E$ as defined in Lemma 2, we have*

$$\max_{a \in \mathcal{A}_t} \mathrm{LCB}_t(a) \geqslant \max_{a \in \mathcal{A}_\tau} \mathrm{LCB}_\tau(a). \tag{52}$$

*for all rounds $t > \tau \geqslant 1$.*

**Proof** Let round index $\tau \geqslant 1$ be arbitrary and let $k \in \operatorname{argmax}_{a \in \mathcal{A}_\tau} \mathrm{LCB}_\tau(a)$. We have $k \in \mathcal{A}_{\tau+1}$ since $\mathrm{UCB}_\tau(k) > \mathrm{LCB}_\tau(k) = \max_{a \in \mathcal{A}_\tau} \mathrm{LCB}_\tau(a)$ by (7) of the anytime quantile bounds. It then follows that

$$\max_{a \in \mathcal{A}_{\tau+1}} \mathrm{LCB}_{\tau+1}(a) \geqslant \mathrm{LCB}_{\tau+1}(k) \geqslant \mathrm{LCB}_\tau(k) = \max_{a \in \mathcal{A}_\tau} \mathrm{LCB}_\tau(j), \tag{53}$$

where the second inequality follows from (7) of the anytime quantile bounds. Applying the argument repeatedly yields the claim for all $t > \tau$. ∎

**Lemma 33 (Elimination of non-satisfying arms)** *Fix an instance $\nu \in \mathcal{E}$, and suppose Algorithm 1 is run with input $(\mathcal{A}, \lambda, \epsilon, q, \delta)$ and parameter $c \geqslant 1$. Let $\mathcal{A}_\epsilon = \mathcal{A}_\epsilon(\nu)$ be as defined in (2) and let the gap $\Delta_k = \Delta_k(\nu, \lambda, \epsilon, c, q)$ be as defined in Definition 10 for each arm $k \in \mathcal{A}$. Consider an arm $k \notin \mathcal{A}_\epsilon$. Under Event $E$ as defined in Lemma 2, when the round index $t$ of Algorithm 1 satisfies $\Delta^{(t)} \leqslant \frac{1}{2}\Delta_k$, we have $k \notin \mathcal{A}_{t+1}$.*

**Proof** If $k \notin \mathcal{A}_t$, then $k \notin \mathcal{A}_{t+1}$ trivially. Therefore, we assume for the rest of the proof that $k \in \mathcal{A}_t$, and we will show that

$$\mathrm{UCB}_t(k) \leqslant \max_{a \in \mathcal{A}_t} \mathrm{LCB}_t(a) \tag{54}$$

or equivalently

$$\mathrm{UCB}_t(k) < \max_{a \in \mathcal{A}_t} \mathrm{LCB}_t(a) + \tilde{\epsilon}, \tag{55}$$

where $\tilde{\epsilon} = \tilde{\epsilon}(\lambda, \epsilon, c)$ is as defined in Lines 1 and 2 of Algorithm 1. Note that these conditions are equivalent because both $\mathrm{UCB}_t(k)$ and $\max_{a \in \mathcal{A}_t} \mathrm{LCB}_t(a)$ are elements of

$$[0, \tilde{\epsilon}, 2\tilde{\epsilon}, \cdots, (n-1)\tilde{\epsilon}, \lambda], \tag{56}$$

which follows from Lines 3, 7, and 11–14 of Algorithm 1.

Since $k \notin \mathcal{A}_\epsilon$, when the round index $t$ satisfies $\Delta^{(t)} \leqslant \frac{1}{2}\Delta_k$ we have

$$\mathrm{UCB}_t(k) < Q_k\big(q + \Delta^{(t)}\big) + \tilde{\epsilon} \leqslant Q_j^+\big(q - \Delta^{(t)}\big) \tag{57}$$

for some arm $j \in \mathcal{A}$ by (9) of the anytime quantile bounds and Definition 10. We now consider two cases: (i) $j \in \mathcal{A}_t$ and (ii) $j \notin \mathcal{A}_t$.

If $j \in \mathcal{A}_t$, we have

$$Q_j^+\big(q - \Delta^{(t)}\big) \leqslant \mathrm{LCB}_t(j) + \tilde{\epsilon} \leqslant \max_{a \in \mathcal{A}_t} \mathrm{LCB}_t(a) + \tilde{\epsilon} \tag{58}$$

by (8) of the anytime quantile bounds and the assumption that $j \in \mathcal{A}_t$. Combining (57) and (58) gives us condition (55) as desired.

If $j \notin \mathcal{A}_t$, then it is eliminated at some round $\tau < t$, i.e., $j \in \mathcal{A}_\tau$ but $j \notin \mathcal{A}_{\tau+1}$. By (7) of the anytime quantile bounds, the definition of active arm set (Line 15 of Algorithm 1) applied to $\mathcal{A}_{\tau+1}$, and the fact that $\max \mathrm{LCB}$ is non-decreasing (Lemma 32), we have

$$Q_j(q) \leqslant \mathrm{UCB}_\tau(j) \leqslant \max_{a \in \mathcal{A}_\tau} \mathrm{LCB}_\tau(a) \leqslant \max_{a \in \mathcal{A}_t} \mathrm{LCB}_t(a). \tag{59}$$

Combining (57), the trivial inequality $Q_j^+\big(q - \Delta^{(t)}\big) \leqslant Q_j(q)$, and (59) yields condition (54) as desired. ∎

**Remark 34** *As seen in the analysis for the case $j \notin \mathcal{A}_t$ above, the property that $\max \mathrm{LCB}$ is non-decreasing (Lemma 32) is crucial in establishing (59). We will see below that the same argument is used again in establishing (68). This property of Lemma 32 itself is a consequence of ensuring $\mathrm{LCB}_t(k)$ is non-decreasing in $t$; see Remark 5.*

**Lemma 35 (While-loop termination)** *Fix an instance $\nu \in \mathcal{E}$, and suppose Algorithm 1 is run with input $(\mathcal{A}, \lambda, \epsilon, q, \delta)$ and parameter $c \geqslant 1$. Let $\mathcal{A}_\epsilon = \mathcal{A}_\epsilon(\nu)$ be as defined in (2) and let the gap $\Delta_k = \Delta_k(\nu, \lambda, \epsilon, c, q)$ be as defined in Definition 10 for each arm $k \in \mathcal{A}$. Under Event E, when the round index $t$ of Algorithm 1 satisfies $\Delta^{(t)} \leqslant \frac{1}{2} \max_{a \in \mathcal{A}_\epsilon} \Delta_a$, Algorithm 1 will terminate in round $t + 1$.*

**Proof** If $\mathcal{A}_{t+1} = \{k^*\}$, then

$$\max_{a \in \mathcal{A}_{t+1} \setminus \{k^*\}} \mathrm{UCB}_t(a) - (c+1)\tilde{\epsilon} = -\infty \leqslant \mathrm{LCB}_t(k^*), \tag{60}$$

and so the algorithm will terminate and return arm $k^*$ in round $t + 1$. Therefore, we assume for the rest of the proof that there exists another arm $a \neq k^*$ such that $a \in \mathcal{A}_{t+1}$.

We first show that the following condition is sufficient to trigger the termination condition of the while-loop (Lines 8–16) of Algorithm 1: There exists an arm $k \in \mathcal{A}_{t+1}$ satisfying

$$\mathrm{LCB}_t(k) \geqslant \max_{a \in \mathcal{A}_{t+1} \backslash \{k\}} Q_a\big(q + \Delta^{(t)}\big) - (c+1)\tilde{\epsilon}. \tag{61}$$

Using (9) of the anytime quantile bound, condition (61) implies that

$$\mathrm{LCB}_t(k) > \max_{a \in \mathcal{A}_{t+1} \backslash \{k\}} \mathrm{UCB}_t(a) - (c+2)\tilde{\epsilon}, \tag{62}$$

which is equivalent to the termination condition

$$\mathrm{LCB}_t(k) \geqslant \max_{a \in \mathcal{A}_{t+1} \backslash \{k\}} \mathrm{UCB}_t(a) - (c+1)\tilde{\epsilon}, \tag{63}$$

where the equivalence follows from an argument similar to the equivalence between (54) and (55).

It remains to pick an arm $k \in \mathcal{A}_{t+1}$ satisfying condition (61). Let arm $j \in \operatorname{argmax}_{a \in \mathcal{A}_\epsilon} \Delta_a$ and consider the following two cases: (i) $j \in \mathcal{A}_{t+1}$ and (ii) $j \notin \mathcal{A}_{t+1}$.

If $j \in \mathcal{A}_{t+1}$, we pick $k = j$. We also pick $T \in \operatorname{argmax}_{\mathcal{A}_\epsilon \subseteq S \subseteq \mathcal{A}} \Delta_k^{(S)}$ to be the set associated to $\Delta_k$ (see Definition 10). Note that every arm that is not in $T$ is a non-satisfying arm since $\mathcal{A}_\epsilon \subseteq T$. Furthermore, every non-satisfying arm that is not in $T$, hence every arm that is not in $T$, is eliminated, which follows from Lemma 33 and

$$\Delta^{(t)} \leqslant \frac{1}{2} \max_{a \in \mathcal{A}_\epsilon} \Delta_a = \frac{1}{2}\Delta_k \leqslant \frac{1}{2} \min_{a \notin T} \Delta_a, \tag{64}$$

where the last inequality follows from applying (13) to $k$ and $T$. Therefore, we have $\mathcal{A}_{t+1} \subseteq T$. It follows that

$$\mathrm{LCB}_t(k) \geqslant Q_k^+\big(q - \Delta^{(t)}\big) - \tilde{\epsilon} \tag{65}$$

$$\geqslant \max_{a \in T \backslash \{k\}} Q_a\big(q + \Delta^{(t)}\big) - (c+1)\tilde{\epsilon} \tag{66}$$

$$\geqslant \max_{a \in \mathcal{A}_{t+1} \backslash \{k\}} Q_a\big(q + \Delta^{(t)}\big) - (c+1)\tilde{\epsilon}, \tag{67}$$

where the first inequality follows from (8) of the anytime quantile bound, the second inequality follows from applying (13) to $k$ and $T$, and the last inequality follows from $\mathcal{A}_{t+1} \subseteq T$.

If $j \notin \mathcal{A}_{t+1}$, we pick an arm $k \in \operatorname{argmax}_{a \in \mathcal{A}_{t+1}} \mathrm{LCB}_t(a)$ arbitrarily. We also pick $T \in \operatorname{argmax}_{\mathcal{A}_\epsilon \subseteq S \subseteq \mathcal{A}} \Delta_k^{(S)}$ and we have $\mathcal{A}_{t+1} \subseteq T$ as in the case above. Furthermore, since $j \notin \mathcal{A}_{t+1}$, we have

$$Q_j^+\big(q - \Delta^{(t)}\big) \leqslant Q_j(q) \leqslant \max_{a \in \mathcal{A}_{t+1}} \mathrm{LCB}_t(a) = \mathrm{LCB}_t(k), \tag{68}$$

where the second inequality follows from an argument similar to (59). It follows that

$$\text{LCB}_t(k) \geqslant Q_j^+\big(q - \Delta^{(t)}\big) \tag{69}$$

$$\geqslant \max_{a \in T \setminus \{j\}} Q_a\big(q + \Delta^{(t)}\big) - c\tilde{\epsilon} \tag{70}$$

$$\geqslant \max_{a \in \mathcal{A}_{t+1} \setminus \{k\}} Q_a\big(q + \Delta^{(t)}\big) - (c+1)\tilde{\epsilon}, \tag{71}$$

where the first inequality follows from (68), the second inequality follows from applying (13) to $j$ and $T$, and the last inequality follows from $\mathcal{A}_{t+1} \subseteq T$. ∎

## Appendix F. Lower Bounds

### F.1. Proof of Theorem 16 (Lower Bound for the Unquantized Variant)

Since we are adapting the instance from (Nikolakakis et al., 2021, Theorem 4), we will omit certain details for brevity and instead will focus on the main differences.

**Proof** [Proof of Theorem 16] Define the following class of distributions parametrized by $w \in (0, q)$:

$$g_w(x) := w\delta(x) + 1 - w, \tag{72}$$

i.e., $g_w$ is a mixture of the Dirac delta function and a uniform distribution on $[0, 1]$. Fix $w, \gamma \in (0, q)$ such that $w + \gamma \leqslant q$. Note that $g_w$ has a higher $q$-quantile than $g_{w+\gamma}$ since

$$G_w^{-1}(q) - G_{w+\gamma}^{-1}(q) = \frac{q - w}{1 - w} - \frac{q - (w + \gamma)}{1 - (w + \gamma)} = \frac{(1 - q)\gamma}{(1 - w)(1 - w - \gamma)} > 0, \tag{73}$$

where $G_w^{-1}$ is the lower quantile function of $g_w$.

We now use (72) to define a set of $K$ instances $\{\nu^{(1)}, \dots, \nu^{(K)}\} \subseteq \mathcal{E}$ for our QMAB problem. Here, each $\nu^{(j)}$ is a different instance of the arm distributions, with $\nu_k^{(j)}$ being the CDF of arm $k$ for instance $j$. Fix $\gamma \in (0, 1/6)$. For $\nu^{(1)}$, we define the arms' PDF by

$$\nu_k^{(1)} := \begin{cases} g_{1/3 - \gamma} & \text{if } k = 1 \\ g_{1/3} & \text{if } k \neq 1. \end{cases} \tag{74}$$

For $j = 2, \dots, K$, we define the arms' PDF of $\nu^{(j)}$ by

$$\nu_k^{(j)} := \begin{cases} g_{1/3 - \gamma} & \text{if } k = 1 \\ g_{1/3 - 2\gamma} & \text{if } k = j \\ g_{1/3} & \text{if } k \neq 1 \text{ or } j. \end{cases} \tag{75}$$

We will use $\nu^{(1)}$ as the "hard instance" in our lower bound. By assumption of our $\epsilon$, we have arm 1 being the unique satisfying arm for $\nu^{(1)}$. Using (16)) and (73), we have

$$\epsilon \leqslant \frac{1}{2}\Big(Q_{k*}^{(1)}(q) - \max_{k \neq k*} Q_k^{(1)}(q)\Big) = \frac{G_{1/3 - \gamma}^{-1}(q) - G_{1/3}^{-1}(q)}{2} = \frac{(1 - q)\gamma}{2(2/3 + \gamma)(2/3)}. \tag{76}$$

This implies that arm $j$ is the unique satisfying arm for $\nu^{(j)}$ for $j = 2, \ldots, K$ since

$$G_{1/3-2\gamma}^{-1}(q) - G_{1/3-\gamma}^{-1}(q) = \frac{(1-q)\gamma}{(2/3+2\gamma)(2/3+\gamma)} = \frac{(1-q)\gamma}{2(2/3+\gamma)(2/3)} \cdot \frac{4/3}{2/3+2\gamma} \geqslant \epsilon, \quad (77)$$

where the inequality follows from (76) and $\gamma \leqslant 1/6$.

To establish the lower bound on the arm pulls for instance $\nu^{(1)}$, we first upper bound the inverse arm gap $\Delta_j^{-1}$ in terms of $\gamma$ for the arms in $\nu^{(1)}$. For arm 1 and each non-satisfying arm $j \neq 1$, we have

$$\Delta_1 \geqslant \Delta_j = \sup \left\{ \Delta \in [0, \min(q, 1-q)] : G_{1/3}^{-1}(q+\Delta) \leqslant G_{1/3-\gamma}^{-1}(q-\Delta) \right\} \quad (78)$$

$$= \min \left\{ \sup \left\{ \Delta \geqslant 0 : \left( \frac{q+\Delta-1/3}{2/3} \right) \leqslant \left( \frac{q-\Delta-1/3+\gamma}{2/3+\gamma} \right) \right\}, q, 1-q \right\} \quad (79)$$

$$= \min \left\{ \frac{(1-q)\gamma}{(4/3+\gamma)}, q, 1-q \right\} \quad (80)$$

$$= \frac{(1-q)\gamma}{(4/3+\gamma)} \quad (81)$$

$$\geqslant \frac{2(1-q)\gamma}{3}, \quad (82)$$

where the first inequality follows from the argument in (14) and the second inequality follows from $\gamma \leqslant 1/6$.

Fix an $(\epsilon, \delta)$-reliable algorithm $\pi$ (see Definition 10), and let $\tau \leqslant \infty$ be the total number of arm pulls by $\pi$ on instance $\nu^{(1)}$. We may assume that $\mathbb{P}_{\nu^{(1)}}[\tau = \infty] = 0$, since otherwise $\mathbb{E}_{\nu^{(1)}}[\tau] = \infty$ and the theorem holds trivially. For each $j \in \{2, \ldots, K\}$, define event $A_j$ to be

$$A_j := \{\pi \text{ terminates and outputs } \hat{k} \neq j\}. \quad (83)$$

By the definition of $(\epsilon, \delta)$-reliability, we must have

$$\mathbb{P}_{\nu^{(j)}}[A_j] \leqslant \delta \text{ for each } j \in \{2, \ldots, K\} \quad \text{and} \quad \mathbb{P}_{\nu^{(1)}}[\tau < \infty \cap \hat{k} \neq 1] \leqslant \delta. \quad (84)$$

Using the assumption $\mathbb{P}_{\nu^{(1)}}[\tau = \infty] = 0$ and the event inclusion $\{\hat{k} = j\} \subseteq \{\hat{k} \neq 1\}$, we have

$$\mathbb{P}_{\nu^{(1)}}[A_j^{\complement}] = \mathbb{P}_{\nu^{(1)}}[\tau = \infty \cup \hat{k} = j] = \mathbb{P}_{\nu^{(1)}}[\hat{k} = j] \leqslant \mathbb{P}_{\nu^{(1)}}[\hat{k} \neq 1] = \mathbb{P}_{\nu^{(1)}}[\tau < \infty \cap \hat{k} \neq 1] \leqslant \delta, \quad (85)$$

and so

$$\mathbb{P}_{\nu^{(1)}}[A_j^{\complement}] + \mathbb{P}_{\nu^{(j)}}[A_j] \leqslant 2\delta \quad \text{for each } j \in \{2, \ldots, K\}. \quad (86)$$

Let $T_j \leqslant \tau$ be the number of times arm $j$ is pulled on $\nu^{(1)}$. For a fixed $j \in \{2, \ldots, K\}$, we have

$$\mathbb{E}_{\nu^{(1)}}[T_j] \geqslant \frac{D_{\mathrm{KL}}(\mathbb{P}_{\nu^{(1)}} \| \mathbb{P}_{\nu^{(j)}})}{12\gamma^2} \geqslant \frac{1}{12\gamma^2} \log\left(\frac{1}{4\delta}\right) \quad (87)$$

where the inequalities follow from (Nikolakakis et al., 2021, Eqn. 29–34). Summing through $j = 2, \ldots, K$ and we have

$$\mathbb{E}_{\nu^{(1)}}[\tau] \geqslant \sum_{j=2}^{K} \mathbb{E}_{\nu^{(1)}}[T_j] \geqslant \sum_{j=2}^{K} \frac{1}{12\gamma^2} \log\left(\frac{1}{4\delta}\right) \geqslant \frac{1}{2} \sum_{j=1}^{K} \frac{1}{12\gamma^2} \log\left(\frac{1}{4\delta}\right), \quad (88)$$

where the last inequality follows from $K \geqslant 2$. Applying the bounds (78)–(82) for each $j$ yields

$$\mathbb{E}_{\nu^{(1)}}[\tau] \geqslant \sum_{j=1}^{K} \frac{(1-q)^2}{27\Delta_j^2} \log\left(\frac{1}{4\delta}\right) = \Omega\left(\sum_{k=1}^{K} \frac{1}{\Delta_k^2} \log\left(\frac{1}{\delta}\right)\right), \tag{89}$$

as desired. ∎

### F.2. Proof of Theorem 17 ($\Omega(\log(\lambda/\epsilon))$ Dependence)

**Proof** [Proof of Theorem 17] Let $\nu$ be a two-arm QMAB instance with deterministic but unknown $q$-quantile rewards $r_1$ and $r_2$ satisfying[10]

$$r_1, r_2 \in [0, 2\epsilon, 4\epsilon, \ldots, \lambda] \quad \text{and} \quad |r_1 - r_2| = 2\epsilon. \tag{90}$$

In this case, the only arm satisfying (2) is the one with the higher $q$-quantile. Since the rewards are deterministic, the QMAB problem is equivalent to finding out which of $r_1$ and $r_2$ is higher.

We consider a modified threshold query setup where the learner receives more information at each iteration: At iteration $t$, the learner decides a threshold $X_t \in [0, \lambda]$, and receives a 2-bit comparison feedback in the form of $(\mathbf{1}(r_1 \leqslant X_t), \mathbf{1}(r_2 \leqslant X_t))$. By design, the number of iterations required under the 2-bit threshold query setup is at most the number of arm pulls required under the 1-bit threshold query setup.

We now establish the lower bound of $\Omega(\log(\lambda/\epsilon))$ on the number of iterations needed to determine which of $r_1$ and $r_2$ is higher for instance $\nu$ under the 2-bit threshold query setup. We first claim that for an algorithm to be $(\epsilon, \delta)$-reliable, the learner has to keep querying until receiving some feedback satisfying

$$(\mathbf{1}(r_1 \leqslant X_t), \mathbf{1}(r_2 \leqslant X_t)) \in \{(0,1), (1,0)\}, \tag{91}$$

which occurs if and only if $X_t \in [\min(r_1, r_2), \max(r_1, r_2)]$. Feedback of the form in (91) is necessary as otherwise instance $\nu$ is indistinguishable from instance $\nu'$ where $r_1$ and $r_2$ are swapped, and the best any algorithm could do is to make a 50/50 guess, which is not $(\epsilon, \delta)$-reliable for $\delta < 0.5$.

To establish the lower bound, we may assume that the learner knows that (90) holds, since extra information can only weaken a lower bound. With this information, instead of picking $X_t$ from the interval $[0, \lambda]$, the learner could pick $X_t$ only from the list $X := [0, 2\epsilon, 4\epsilon, \ldots, \lambda]$ without loss of generality (any other choices would have a corresponding equivalent choice in this set). As there is exactly one $x \in X$ that would lead to feedback of the form in (91), we need to identify one of $|X|$ possible outcomes, which amounts to learning $\log_2 |X|$ bits. Since each threshold query gives a 2-bit feedback, the number of threshold queries/iterations needed in the worst case is $\Omega(\log(|X|)) = \Omega(\log(\lambda/\epsilon))$. ∎

---

10. For ease of analysis, we assume $\lambda$ is an integer multiple of $2\epsilon$.

## Appendix G.  Proof of Theorem 21 and Corollary 24 (Solvable Instances)

We first state a useful lemma for Theorem 21.

**Lemma 36** *Let $\lambda, \epsilon, c$, and $q$ be given, and let $\tilde{\epsilon} = \tilde{\epsilon}(\lambda, \epsilon, c)$ be as defined in Algorithm 1. Suppose that $\nu \in \mathcal{E}$ is an instance with gap $\Delta(\nu, \lambda, \epsilon, c, q) = 0$ and let $\eta_0 = \eta_0(\nu) > 0$ be the constant given in the assumption in Theorem 21. Then, for each arm $k \in \mathcal{A}_{c\tilde{\epsilon}}(\nu)$ and each $\eta \in (0, \eta_0)$, there exists another instance $\nu' \in \mathcal{E}$ satisfying the following:*

- *There exists an arm $a \in \mathcal{A} \backslash \{k\}$ such that instances $\nu$ and $\nu'$ are identical for all arms in $\mathcal{A} \backslash \{a, k\}$;*
- *$d_{\mathrm{TV}}(F_a, G_a) \leqslant \eta$ and $d_{\mathrm{TV}}(F_k, G_k) \leqslant \eta$, where $F_{(\cdot)}$ and $G_{(\cdot)}$ represent the arm distributions for instances $\nu$ and $\nu'$ respectively;*
- *$k \notin \mathcal{A}_{c\tilde{\epsilon}}(\nu')$, i.e., under relaxation parameter $c\tilde{\epsilon}$, arm $k$ is not a satisfying arm for instance $\nu'$.*

**Proof** Let $\nu \in \mathcal{E}$ be an instance with gap $\Delta(\nu, \lambda, \epsilon, c, q) = 0$. For each arm $k \in \mathcal{A}_\epsilon(\nu)$, we have $\Delta_k^{(\mathcal{A})} = 0$ by Definition 10 since $0 \leqslant \Delta_k^{(\mathcal{A})} \leqslant \Delta_k \leqslant \Delta = 0$. Applying (13) with set $S = \mathcal{A}$ yields:

$$\text{for each } k \in \mathcal{A}_\epsilon(\nu) \text{ and each } \eta > 0, \text{ there exists } a \neq k \text{ such that } Q_k^+(q-\eta) < Q_a(q+\eta) - c\tilde{\epsilon}. \quad (92)$$

Fix an arm $k \in \mathcal{A}_{c\tilde{\epsilon}}(\nu)$ and $\eta \in (0, \eta_0)$. Since $c\tilde{\epsilon} \leqslant \epsilon$ (see calculation in (41)–(42)), we have $\mathcal{A}_{c\tilde{\epsilon}}(\nu) \subseteq \mathcal{A}_\epsilon(\nu)$, and hence $k \in \mathcal{A}_\epsilon(\nu)$. It follows from (92) that there exists some arm $a \neq k$ that

$$Q_k^+(q - \eta) < Q_a(q + \eta) - c\tilde{\epsilon}. \quad (93)$$

We now construct instance $\nu'$ such that $\nu$ and $\nu'$ have identical distributions for all arms in $\mathcal{A} \backslash \{a, k\}$, while $F_a$ and $F_k$ are being replaced with $G_a$ and $G_k$ defined as follows:

1. $G_a$ is any distribution obtained by moving $\eta$-probability mass from the interval $(-\infty, Q_a(q))$ to the point $Q_a(q + 2\eta)$;

2. $G_k$ is any distribution obtained by moving $\eta$-probability mass from the interval $(Q_k(q), \infty)$ to the point $Q_k(q - 2\eta)$.

Under these definitions and the assumption on $\eta_0$ in Theorem 21, we can readily verify that

$$(G_k)^{-1}(q) = Q_k(q - \eta) \in [0, \lambda] \quad \text{and} \quad (G_a)^{-1}(q) = Q_a(q + \eta) \in [0, \lambda] \quad (94)$$

and

$$d_{\mathrm{TV}}(F_k, G_k) = d_{\mathrm{TV}}(F_a, G_a) = \eta. \quad (95)$$

Finally, combining (93) and (94) yields

$$(G_k)^{-1}(q) < (G_a)^{-1}(q) - c\tilde{\epsilon}, \quad (96)$$

which implies $k \notin \mathcal{A}_{c\tilde{\epsilon}}(\nu')$. By construction, $\nu'$ satisfies all three properties as desired. ■

**Remark 37** *We can obtain a "limiting" version of Lemma 36 in which we replace the gap $\Delta(\nu, \lambda, \epsilon, c, q)$ by $\Delta(\nu, \epsilon, q)$ as defined in Corollary 24 and the satisfying arm set $\mathcal{A}_{c\tilde{\epsilon}}(\cdot)$ by $\mathcal{A}_\epsilon(\cdot)$. The proof is essentially identical. We construct instance $\nu'$ in a similar manner as above to satisfy the first two properties in the statement of Lemma 36. The last property $(k \notin \mathcal{A}_\epsilon(\nu'))$ then follows from the definition of the limit gap $\Delta_k(\nu, \epsilon, q)$ as defined in (43), which allows us to replace the $c\tilde{\epsilon}$ terms in (92), (93), and (96) by $\epsilon$.*

We proceed to prove Theorem 21.

**Proof** [Proof of Theorem 21] Assume for contradiction that there exists some instance $\nu \in \mathcal{E}$ satisfies $\Delta(\nu, \lambda, \epsilon, c, q) = 0$, but is $c\tilde{\epsilon}$-solvable. Fix a $\delta \in (0, 1)$ satisfying

$$\delta < \frac{1}{2 + 2|\mathcal{A}|}. \tag{97}$$

By Definition 18, there exists a $(c\tilde{\epsilon}, \delta)$-reliable algorithm such that

$$\mathbb{P}_\nu[\tau < \infty \cap \hat{k} \in \mathcal{A}_{c\tilde{\epsilon}}(\nu)] \geq 1 - \delta. \tag{98}$$

In general the condition $\tau < \infty$ may not imply a *uniform* upper bound on $\tau$; we handle this by relaxing the probability from $1 - \delta$ to $1 - 2\delta$, such that there exists some $\tau_{\max} < \infty$ satisfying

$$\mathbb{P}_\nu[\hat{k} \in \mathcal{A}_{c\tilde{\epsilon}}(\nu) \cap \tau \leq \tau_{\max}] \geq 1 - 2\delta. \tag{99}$$

From this, we claim that there exists an arm $k_\nu \in \mathcal{A}_{c\tilde{\epsilon}}(\nu)$ such that

$$\mathbb{P}_\nu[\hat{k} = k_\nu \cap \tau \leq \tau_{\max}] \geq \frac{1 - 2\delta}{|\mathcal{A}|}. \tag{100}$$

Indeed, if this were not the case, then summing these probabilities over elements in $\mathcal{A}_\epsilon(\nu)$ would produce a total below $1 - 2\delta$, which would contradict (99).

Let $P_{\tau_{\max}}^{(\nu)}$ be the joint distribution on the $|\mathcal{A}| \times \tau_{\max}$ matrix of unquantized rewards: the $(i, j)$-th entry of this matrix contains the $j$-th unquantized reward for arm $i$ under instance $\nu$. Under the event $\tau \leq \tau_{\max}$, the algorithm's output does not depend on any rewards beyond those appearing in this matrix. In other words, the output $\hat{k}$ is a (possibly randomized) function of this matrix.

By picking $\eta > 0$ to be sufficiently small in Lemma 36, we can find an instance $\nu' \in \mathcal{E}$ such that $k_\nu \notin \mathcal{A}_{c\tilde{\epsilon}}(\nu')$ and

$$d_{\mathrm{TV}}\big(P_{\tau_{\max}}^{(\nu)}, P_{\tau_{\max}}^{(\nu')}\big) \leq \delta. \tag{101}$$

Here, $P_{\tau_{\max}}^{(\nu')}$ is defined similarly to $P_{\tau_{\max}}^{(\nu)}$, but for instance $\nu'$. Since the output $\hat{k}$ is a (possibly randomized) function of the matrix defining $P_{\tau_{\max}}^{(\cdot)}$, we have

$$d_{\mathrm{TV}}\big(\mathbb{P}_\nu, \mathbb{P}_{\nu'}\big) \leq d_{\mathrm{TV}}\big(P_{\tau_{\max}}^{(\nu)}, P_{\tau_{\max}}^{(\nu')}\big) \leq \delta \tag{102}$$

by the data processing inequality for $f$-divergence (Polyanskiy and Wu, 2025, Theorem 7.4). Using the definition $d_{\mathrm{TV}}(P, Q) = \sup_A |P(A) - Q(A)|$, and applying (102), (100), (97), we obtain

$$\mathbb{P}_{\nu'}[\hat{k} = k_\nu \cap \tau \leq \tau_{\max}] \geq \mathbb{P}_\nu[\hat{k} = k_\nu \cap \tau \leq \tau_{\max}] - d_{\mathrm{TV}}\big(\mathbb{P}_\nu, \mathbb{P}_{\nu'}\big) \geq \frac{1 - 2\delta}{|\mathcal{A}|} - \delta > \delta. \tag{103}$$

Since $k_\nu \notin \mathcal{A}_{c\tilde{\epsilon}}(\nu')$, this means that the algorithm is *not* $(c\tilde{\epsilon}, \delta)$-reliable (see Definition 10), we have arrived at the desired contradiction. ∎

Corollary 24 can be proved similarly by using the "limiting" version of Lemma 36 (see Remark 37).

## Appendix H. Details on Remark 13 (Improved Gap Definition)

### H.1. Modified Arm Gaps

We first state the modified gap definition explicitly by replacing $Q^+_{(\cdot)}(q - \Delta)$ and $Q_{(\cdot)}(q + \Delta)$ in Definition 10 with $\max\{0, Q^+_{(\cdot)}(q - \Delta)\}$ and $\min\{\lambda, Q_{(\cdot)}(q + \Delta)\}$ respectively, and provide an instance that has a positive modified gap but zero gap under the original definition.

**Definition 38 (Modified arm gaps)** *Fix an instance $\nu \in \mathcal{E}$. Let $\tilde{\epsilon}$ and $\mathcal{A}_\epsilon$ be as in Definition 10. For each arm $k \in \mathcal{A}$, we define the improved gap $\tilde{\Delta}_k = \tilde{\Delta}_k(\nu, \lambda, \epsilon, c, q) \in [0, \min(q, 1 - q)]$ as follows:*

- *If $k \notin \mathcal{A}_\epsilon$, then $\tilde{\Delta}_k$ is defined as*

$$\sup\left\{\Delta \in [0, \min(q, 1 - q)] : \min\{\lambda, Q_k(q + \Delta)\} \leqslant \max_{a \in \mathcal{A}}\left\{\max\left\{0, Q^+_a(q - \Delta)\right\} - \tilde{\epsilon}\right\}\right\} \tag{104}$$

- *If $k \in \mathcal{A}_\epsilon$, then we define $\tilde{\Delta}_k = \max_{\mathcal{A}_\epsilon \subseteq S \subseteq \mathcal{A}} \tilde{\Delta}_k^{(S)}$, where*

$$\tilde{\Delta}_k^{(S)} = \sup\left\{\Delta \in \left[0, \min_{a \notin S} \tilde{\Delta}_a\right] : \max\{0, Q^+_k(q - \Delta)\} \geqslant \max_{a \in S \setminus \{k\}} \min\{\lambda, Q_a(q + \Delta)\} - c\tilde{\epsilon}\right\} \tag{105}$$

*for each subset $S$ satisfying $\mathcal{A}_\epsilon \subseteq S \subseteq \mathcal{A}$.*

*We use the convention that the minimum (resp. maximum) of an empty set is $\infty$ (resp. $-\infty$).*

**Remark 39 (Intuition on the modified arm gap)** *Fix an instance $\nu = (F_k) \in \mathcal{E}$. An interpretation of this modified gap is that $\tilde{\Delta}_k(\nu, \lambda, \epsilon, c, q) = \Delta_k(\mathrm{clipped}(\nu), \lambda, \epsilon, c, q)$, where $\mathrm{clipped}(\nu) = (\tilde{F}_k) \in \mathcal{E}$ is the instance with all distributions supported on $[0, \lambda]$ defined by*

$$\tilde{F}_k(x) = \begin{cases} 0 & \text{for } x < 0 \\ F_k(x) & \text{for } 0 \leqslant x < \lambda \quad \text{for each } k \in \mathcal{A}. \\ 1 & \text{for } x > \lambda \end{cases} \tag{106}$$

*That is, $\tilde{F}_k$ is obtained from $F_k$ by moving all mass below 0 to 0, and all mass above $\lambda$ to $\lambda$. Note that an algorithm could be designed to clip rewards in this way, but our improved upper bound in Theorem 40 below applies even when Algorithm 1 is run without change.*

It is straightforward to verify that the modified gap is at least as large as the unmodified gap (Definition 10), i.e., $\tilde{\Delta} \geqslant \Delta$. We provide an example of bandit instance that has positive gap under the modified definition but is zero using the unmodified definition. Consider $q = 1/2$, let $\lambda \geqslant 2\epsilon > 0$, and consider two arms $\mathcal{A} = \{1, 2\}$ with an identical CDF as follows:

$$F_1(x) = F_2(x) = \begin{cases} 0 & \text{for } x < \lambda - \epsilon/3 \\ 0.5 & \text{for } \lambda - \epsilon/3 \leqslant x < 2\lambda \,, \\ 1 & \text{for } x \geqslant 2\lambda \end{cases} \tag{107}$$

and so both arms are satisying, i.e., $\mathcal{A}_\epsilon = \mathcal{A}$. Note that for any $\Delta > 0$, we have

$$Q_2^+(0.5 - \Delta) = \lambda - \epsilon/3 < 2\lambda - \epsilon \leqslant 2\lambda - c\tilde{\epsilon} = Q_1(0.5 + \Delta) - c\tilde{\epsilon}, \tag{108}$$

where the second inequality follows from the discussion in (41)–(42). It follows that

$$\Delta_2 = \Delta_2^{\mathcal{A}} = \sup\left\{\Delta \in [0, 0.5] : Q_2^+(0.5 - \Delta) \geqslant Q_1(0.5 + \Delta) - c\tilde{\epsilon}\right\} = 0 \tag{109}$$

under the original gap definition. By symmetry, we also have $\Delta_1 = 0$. However, under the modified definition, we have

$$\tilde{\Delta}_2 = \tilde{\Delta}_2^{\mathcal{A}} = \sup\left\{\Delta \in [0, 0.5] : \max\{0, \lambda - \epsilon/2\} \geqslant \min\{\lambda, 2\lambda\} - c\tilde{\epsilon}\right\} \tag{110}$$

$$= \sup\left\{\Delta \in [0, 0.5] : \lambda - \epsilon/3 \geqslant \lambda - c\tilde{\epsilon}\right\} \tag{111}$$

$$= 0.5, \tag{112}$$

where the last inequality follows since $c\tilde{\epsilon} \geqslant \epsilon/3$ for any $c \geqslant 1$ (see the calculation in Remark 23).

## H.2. Improved Upper Bound

With the modified gap definition, we obtain the following improved upper bound.

**Theorem 40 (Improved upper bound)** *Fix an instance $\nu \in \mathcal{E}$, and suppose Algorithm 1 is run with input $(\mathcal{A}, \lambda, \epsilon, q, \delta)$ and parameter $c \geqslant 1$. Let $\mathcal{A}_\epsilon(\nu)$ be as defined in (2) and let the gap $\tilde{\Delta}_k = \tilde{\Delta}_k(\nu, \lambda, \epsilon, c, q)$ be as defined in Definition 38 for each arm $k \in \mathcal{A}$. Under Event E as defined in Lemma 2, the total number of arm pulls is upper bounded by*

$$O\left(\left(\sum_{k \in \mathcal{A}} \frac{1}{\max\left(\tilde{\Delta}_k, \tilde{\Delta}\right)^2} \cdot \left(\log\left(\frac{1}{\delta}\right) + \log\left(\frac{1}{\max\left(\tilde{\Delta}_k, \tilde{\Delta}\right)}\right) + \log\left(\frac{c\lambda K}{\epsilon}\right)\right)\right)\right), \tag{113}$$

*where $\tilde{\Delta} = \tilde{\Delta}(\nu, \lambda, \epsilon, c, q) = \max_{a \in \mathcal{A}_\epsilon(\nu)} \tilde{\Delta}_a$.*

The proof is essentially identical to the proof of Theorem 14, but requires tightening of (8) and (9) of anytime quantile bound to

$$\max\{0, Q_k^+(q - \Delta^{(t)})\} \leqslant \text{LCB}_t(k) + \tilde{\epsilon} \tag{114}$$

and

$$\mathrm{UCB}_t(k) < \min\{\lambda, Q_k(q + \Delta^{(t)})\} + \tilde{\epsilon} \tag{115}$$

respectively. Note that the two new bounds (114) and (115) can be verified easily using the properties that $\mathrm{LCB}_t(k) \geqslant 0$ and $\mathrm{UCB}_t(k) \leqslant \lambda$ (see Lines 7, 12, and 14 of Algorithm 1), as well as the established bounds (8) and (9).

## H.3. Removing the Assumption in Theorem 21 (Unsolvability)

The assumption involving $\eta_0$ in Theorem 21 is included to ensure that both $(G_k)^{-1}(q) = Q_k(q - \eta)$ and $(G_a)^{-1}(q) = Q_a(q + \eta)$ are in $[0, \lambda]$ in the proof of Lemma 36, so that the constructed instance $\nu'$ satisfies $\nu' \in \mathcal{E}$. As mentioned in Remark 22, the assumption can be removed if we use the modified gap instead; formally, we have the following.

**Theorem 41 (Zero gap is unsolvable – assumption-free version)** *Let $\lambda, \epsilon, c$, and $q$ be fixed, and let $\tilde{\epsilon} = \tilde{\epsilon}(\lambda, \epsilon, c)$ be as defined in Algorithm 1. Let $\tilde{\Delta} = \tilde{\Delta}(\nu, \lambda, \epsilon, c, q)$ be as defined in Theorem 40. If an instance $\nu \in \mathcal{E}$ satisfies $\tilde{\Delta} = 0$, then $\nu$ is $c\tilde{\epsilon}$-unsolvable.*

The proof is essentially identical to the proof of Theorem 21, and requires only some straightforward modifications in Lemma 36. Specifically, under the new gap definition, (93) would be replaced by

$$\max\{0, Q_k^+(q - \eta)\} < \min\{\lambda, Q_a(q + \eta)\} - c\tilde{\epsilon}\} \tag{116}$$

We then construct instance $\nu'$ in a similar manner to the proof of Lemma 36, but the definitions of $G_a$ and $G_k$ modified to include clipping:

1. $G_a$ is any distribution obtained by moving $\eta$-probability mass from the interval $(-\infty, Q_a(q))$ to the point $\min\{\lambda, Q_a(q + 2\eta)\}$;

2. $G_k$ is any distribution obtained by moving $\eta$-probability mass from the interval $(Q_k(q), \infty)$ to the point $\max\{0, Q_k(q - 2\eta)\}$.

It now follows that

$$(G_k)^{-1}(q) = \max\{0, Q_k(q - \eta)\} \in [0, \lambda] \tag{117}$$

and

$$(G_a)^{-1}(q) = \min\{\lambda, Q_a(q + \eta)\} \in [0, \lambda], \tag{118}$$

and hence $\nu' \in \mathcal{E}$ as desired.

