# OpenReview forum: "Quantile Multi-Armed Bandits with 1-bit Feedback"
_algorithmiclearningtheory.org/ALT/2025/Conference — ALT 2025_

### Official Review · Reviewer_7zKk · 2024-11-06
**Quantile Multi-Armed Bandits with 1-bit Feedback**

**Rating:** 7
**Confidence:** 4

**Review:**

This paper considers the problem of best-arm identification in
multi-armed bandit problems where the best arm is defined by the one
whose reward has the highest quantile (for some pre-specified
level). The authors study a variant of the problem when the player
only receives one bit of information from the reward of the selected arm.
More precisely, at each round, the player can ask a question about the
magnitude of the reward and receives a yes/no answer. To the best of
my knowledge, this exact problem has not been studied before.

The main results establish nearly matching upper and lower bounds for
the smallest number of arm pulls that guarantees (approximate) best
arm identification. The techniques are natural and well-known in the
area, though there are quite a few nontrivial details that the authors
need to sort out. The paper is well and clearly written. This is a
solid contribution and I support acceptance at ALT.

Some remarks and questions:

- is it necessary to assume that the range \lambda is known to the player?

- p.6, l.10: \tilde\epsilon is undefined at this point.

- the input parameter c in Algorithm 1 unnatural (as it is not part of
   the problem) and requires more discussion.

-p.11, l.6 and elsewhere: there are two conflicting definitions of the quantity
    \Delta_k. The one in (13) depends on the parameter c while the one appearing
    in Theorem 13 doesn't. This needs some fixing and clarification.

**Paper Award:**

No

---

> ### Author Response · Authors · 2024-11-22
> **Response to Reviewer 7zKk**
>
> We thank the reviewer for the positive evaluation and the useful remarks. Our responses are as follows.
>
> **Assumption on the range of $\lambda$:** We assume that the desired quantile lies in $[0,\lambda]$, meaning that any “over-estimate” of $\lambda$ is still sufficient.  Even if we over-estimate by a large amount, the impact is minimal because the dependence of the regret on $\lambda$ is only logarithmic.  See ‘Concern 3’ in the response to Reviewer exKE for related discussion.
>
> **Input parameter c in Algorithm 1:** This parameter is currently briefly discussed on p.7, right before Section 3.2. We will further add an explicit forward-reference to Section 5 (particularly Remark 21) on how picking a larger $c$ in the algorithm may be beneficial, as Section 5 indeed explores this in much more detail.
>
> **$\tilde{\epsilon}$ being undefined yet on page 6:** We will add a forward reference to Lines 1-2 of Algorithm 1 (page 7).
>
> **$\Delta_k$ in Theorem 13:** It is mentioned in Theorem 13 that the $\Delta_k$ used (in the theorem) is the one defined in (18), which corresponds to $c \to \infty$.  For the final version, we will review the presentation in case this needs to be highlighted more.

---

### Official Review · Reviewer_exKE · 2024-11-09
**Review of "Quantile Multi-Armed Bandits with 1-bit Feedback"**

**Rating:** 5
**Confidence:** 5

**Review:**

This paper addresses a variant of the best-arm identification (BAI) problem where the best arm is defined as the one with the highest q-quantile reward, with a key communication constraint: feedback from the sensor to the learner is limited to a single binary (1-bit) response. Specifically, the learner sends threshold queries of the form, "Is the reward smaller than a given threshold?" to the sensor, which responds with a single bit indicating whether the threshold was exceeded. The authors relaxed the problem into identifying ϵ-best arm(s). A crucial assumption in the paper is that the q-quantiles of all arms are bounded by a known constant, an assumption that may be questionable in cases where rewards are unbounded.

The authors propose an algorithm capable of identifying the best arm with a specified confidence level and derive an upper bound on the sample complexity. This upper bound consists of three main terms: the first matches a lower bound for the case without communication constraints up to a logarithmic factor, the second depends on the logarithm of the inverse optimality gap, and the third is a logarithmic term influenced by a tuning parameter, the number of arms, the quantile bound, and the relaxation margin ϵ. The dependence on the first term suggests that the 1-bit restriction does not significantly affect the algorithm’s performance in the "low error probability/high confidence level" regime. Additionally, the authors present a hardness result, showing that the third term is unavoidable in the worst case if the threshold query model is used. The authors conclude with a discussion on solvable instances based on the sub-optimality gap, modified to account for ϵ-relaxation, along with a review in the appendix on how traditional sub-optimality gaps are less effective under this relaxation setting.

Major concerns:
1. Inverse-Square-Based Problem Complexity: I recognize that bounds of this type are common in the BAI literature (e.g., Mannor and Tsitsiklis, 2004; Nikolakakis et al., 2021 for QMAB). However, I am concerned about the practicality of such bounds in this context. For instance, a closer examination of Theorem 13 reveals a 1/27 factor hidden in the big-O notation. Similarly, while the authors did not show it explicitly, I suspect large constants are also present in Theorem 11. Such large hidden constants can undermine the practical relevance of improvements in sub-optimality gaps, as they reduce the actual efficiency gains compared to prior work.
2. Focus on Sharp, Constant Optimality in Recent BAI Advances: Recent developments in BAI have emphasized not only order optimality but also sharp, constant optimality, particularly in the asymptotic low-error probability regime (e.g., Garivier and Kaufmann, 2016, and follow-up works). These works introduce a clear and explicit form for problem complexity measures instead of relying on big-O notation and inverse-square-gaps, along with a discussion on the connection to gap-based complexity measures. I find this approach more informative and practical, as it provides more accurate insights into algorithm efficiency.
3. Strong Assumption on Bounded Quantiles: The assumption that quantiles are bounded appears overly restrictive and seems introduced primarily for analytical convenience. For instance, in parameterized settings (e.g., normal distributions), high quantile values naturally arise without necessarily increasing problem complexity. Such a restriction appears especially limiting as the bound must be directly accessible to the algorithm.
4. Lack of Clarity in Algorithm 2 Description: The description of Algorithm 2 lacks clarity. For instance, Line 1 iterates over t from 1 to O(1/\Delta^2 \log(n/\delta)). Using big-O notation within an algorithm is problematic - how would this bound be specified in practice, especially given its direct relationship to sample complexity? Furthermore, including more details from Gretta and Price (2024) would enhance completeness and credibility, as many details critical to implementation are currently hidden.
5. No Numerical Evaluation of the Proposed Algorithm: The absence of numerical results raises concerns about the practical applicability of the proposed algorithm. Numerical experiments would help illustrate the algorithm's practical performance, especially compared to existing algorithms. Given the claim that "restricting to 1-bit feedback has a minimal impact," empirical validation would strengthen this claim by demonstrating the algorithm’s comparative performance under full feedback conditions.

Additional concerns:
1. Complexity of Gap Definition: The gap definition in (13) is significantly more complex than in prior work. While using the upper quantile level may improve gaps in cases of discrete support, the improvement for continuous distributions (e.g., normal) is unclear. How does considering all subsets S improve the gap? Simple examples illustrating this would be helpful. Also, a discussion on when these gaps are positive would clarify the practical implications of this definition.
2. Implications of ϵ-Relaxation: The use of ϵ-relaxation changes the problem fundamentally from standard quantile-BAI. Specifically, the potential for multiple correct answers (Degenne and Koolen, 2019) introduces additional challenges, as this scenario is known to be more complex than conventional BAI. For QMAB, could exact quantile-BAI potentially be easier than the ϵ-relaxed version? Furthermore, the paper lacks a discussion on setting an appropriate ϵ value, which is a crucial factor in applications of ϵ-relaxation.

**Paper Award:**

No

---

> ### Author Response · Authors · 2024-11-22
> **Response to Reviewer exKE**
>
> We thank the reviewer for the feedback. We will address the concerns point by point.
>
> **Major concerns 1. and 2. (constant factors hidden in Big-O):** It is standard practice for theory-oriented papers to first establish scaling laws before seeking precise constants. As we are the first to study this 1-bit QMAB problem, we believe this is justified.  In fact, even the study of unquantized QMAB is still in its infancy compared to other bandit settings (e.g., to our knowledge there is only one existing lower bound).
>
>
> **Major concern 3 (bounded quantile):** We first emphasize that having a $q$-quantile being bounded by a known constant does not restrict the rewards from being unbounded. In fact, it is a much milder assumption compared to requiring the rewards being bounded. The bounds on the quantiles can be very crude since the dependence on $\lambda$ is logarithmic in the final bound. Moreover, it is typically unavoidable for 1-bit statistical estimation tasks to assume that the parameter of interest lies on some bounded interval:  In our setup, our lower bound for the threshold query model indicates that $\Omega(\log \frac{\lambda}{\epsilon})$ arm pulls are needed even in the case of deterministic two-arm bandits, indicating the necessity of bounded $\lambda$.
>
> **Major concern 4 (clarity in description for Algorithm 2):** We are happy to modify Algorithm 2 to state the number of loops explicitly with the constant as given in Gretta and Price (2024), and include more details in that section to enhance the completeness.  This will only require minor edits.
>
> **Major concern 5 (numerical experiments)** Continuing from our response to concerns 1 & 2, we believe it is natural for an initial paper on a topic to leave important questions open, such as constant factors (concerns 1 & 2) and experimental validation (concern 5).  Despite these remaining questions, we believe that what we have provided is comfortably sufficient for publication.
>
> **Additional concern 1 (gap definition):** Our gap definition in (13) generalizes and slightly improves those defined in the unquantized QMAB literature.  As an example, we showed in Appendix G that the gap can be positive under our definition while being zero using the existing definitions. We will add a cross-reference to Appendix G after our gap definition to highlight this advantage more clearly. Perhaps more importantly, we showed in Section 5 that essentially all solvable bandit instances have a positive gap under our gap definition, thus providing strong support for our proposed gaps.
>
> Regarding the question on the use of upper quantile function, indeed the gap improvement would happen only in cases of discrete reward distributions, since the upper and lower quantile functions for continuous distributions are the same by definition.  Regarding the question on taking the maximum over all $S$, it is stated in Remark 8 that the quantity  $\Delta_k^{(S)}$ for an arbitrary fixed subset $S$ captures how much better a satisfying arm $k$ is than the arms in $S$ (up to epsilon). When arm $k$ is sufficiently pulled relative to this quantity, the algorithm would be able to conclude that arm $k$ satisfies the relaxed goal, and terminates. Since the conclusion derived from any such set $S$ is valid, we define the gap based on the set $S$ that maximizes $\Delta_k^{(S)}$.
>
>
>
> **Additional concern 2 ($\epsilon$-relaxation):** While the $\epsilon$-relaxation is not needed in the unquantized case, achieving this relaxed goal for 1-bit QMAB under the threshold query model requires $\Omega(\log\frac{\lambda}{\epsilon})$ arm pulls, showing the necessity of this relaxation (see Section 4.2). The choice of $\epsilon$ should be set by the user of the algorithm, since it is dependent on the task/application under consideration.  Leaving it as a free parameter aligns with other well-established bandit papers involving relaxations (e.g., see “The True Sample Complexity of Identifying Good Arms”, Katz-Samuels and Jamieson 2020, and the references therein).

---

### Official Review · Reviewer_FJeN · 2024-11-09
**Updated review**

**Rating:** 5
**Confidence:** 4

**Review:**

The paper studies the following best-arm identification problem. There are n arms with stochastic rewards drawn from unknown distribution $F_i$ for each arm $i$. The learner wants to identify the arm with the largest quantile $F^{-1}_i(q)$ for a given $q$. When pulling an arm, instead of knowing the reward, the learner only observes a single-bit information. In this paper, the single-bit information is from a threshold query, and in each round the learner can observe whether the value of the arm is greater than the threshold proposed by the learner. The paper proposes an algorithm that has asymptotically optimal query complexity.

Evaluation:
The problem studied by the paper is definitely a generalization of the best-arm identification problem of quantile bandit with full value feedback. However, the algorithm proposed by the paper seems straightforward (i.e. run noisy binary search algorithms and maintain the confidence interval around the target quantile of each arm simultaneously), and the idea of using noisy binary search to learn the quantile of a distribution is pretty common, especially in the dynamic pricing literature that always considers pricing (i.e. threshold) queries. Overall, although the authors have made non-trivial technical effort to define the proper terms such that the query complexity is well-defined, I do not find the algorithm and analysis surprising or containing any interesting enough new intuitions.

The line of work that should be cited: learning distribution with threshold/pricing queries has received some recent attention in the learning and game theory area, dating back to “The value of knowing a demand curve: Bounds on regret for online posted-price auctions” (Kleinberg Leighton FOCS’03), with some recent follow-up papers using threshold queries to learn distributions, including: “Pricing Query Complexity of Revenue Maximization” (Paes Leme et al. SODA’23, which includes an algorithm that uses noisy binary search with confidence interval to learn a quantile of a single distribution), “Non-Stochastic CDF Estimation Using Threshold Queries” (Okoroafor et al. SODA’23), “Description Complexity of Regular Distributions” (Paes Leme et al. EC’23), and so on.

**Paper Award:**

No

---

> ### Author Response · Authors · 2024-11-22
> **Response to Reviewer FJeN**
>
> We thank the reviewer for the feedback.
>
> Regarding the extra references, we very much appreciate the reviewer for pointing out the relevant work in pricing query literature, such as "Pricing Query Complexity of Revenue Maximization" by Paes Leme et al. We will include these references and discuss them in the related work section, and we will further highlight the similarity to Paes Leme et al.’s work in Appendix A, as it appears to be the most relevant.
>
> Regarding the novelty, we note that while existing ideas from literature in bandits or quantile estimation (of a single distribution) with threshold/pricing queries may provide useful context, they do not readily translate into our main contributions (with nearly matching upper and lower bounds).  In fact, the analysis of the estimation subroutine only occupies 1.5 pages in Appendix A, so constitutes a relatively minor part of the paper.
>
> Elaborating on this, we believe that there are several details (as pointed out by Reviewer 7zKk) that we need to sort out that are not obvious a priori, including the following:
>
> 1.) We needed to carefully select the performance goal via an $\epsilon$-relaxation, which is not needed in the unquantized case. As discussed in Section 4.2, achieving the relaxed goal under the threshold query model requires $\Omega(\log\frac{\lambda}{\epsilon})$ arm pulls even in the case of deterministic two-arm bandits, which shows the necessity of the relaxation.
>
> 2.) We needed to identify the appropriate gap definition to ensure that we are able to solve essentially all instances that are solvable (see Section 5).  This required a number of subtle design choices including the parameter $c$ (discussed in Remark 9), the optimization over $S$ (see (13)), and the use of upper and lower quantile functions (see Remark 10).  We strengthen existing gaps for the unquantized setting in the sense that those are recovered by considering $c \to \infty$, $S = \\{k^*\\}$, and only lower quantile functions.  See also Appendix G for an example with a strict improvement.
>
> 3.) Removing the dependence on the gap of the best arm required a more complicated analysis (see Remarks 10, 27, and 4), and we appear to be the first to do this for an elimination-based algorithm for QMAB. One existing QMAB paper (“Sequential estimation of quantiles with applications to A/B testing and best-arm identification” by Howard and Ramdas) also managed to similarly remove this dependency, but they use an LUCB-style algorithm whose analysis is very different.
>
> 4.) Overall, the above considerations ultimately required a careful and often delicate analysis for all of our results: The upper bounds (Section 3 & Appendices B–D), lower bounds (Section 4 & Appendix E), and solvable instances (Section 5 & Appendix F).

---

### Author Rebuttal · Authors · 2024-11-22

We sincerely thank all reviewers for their valuable feedback. We have responded individually to each reviewer using the “Official Comment” feature, and we welcome any further discussions or requests for clarification.

---

### Meta-Review · Area_Chair_bAxP · 2024-12-08

**Recommendation:** Accept
**Confidence:** 4

**Metareview:**

The paper introduces the problem of best-arm identification (BAI) with the objective of finding the arm whose reward distribution has the highest $q$-quantile for a specified $q$. Unlike some prior works on quantile BAI, the feedback model is threshold queries, where pulling an arm yields only one bit of the information: whether or not the reward was above or below a specified threshold.

An algorithm is presented which solves this problem within $\varepsilon$, i.e. with probability close to 1, it outputs an arm whose $q$-quantile is within $\varepsilon$ of the best. The number of pulls of each suboptimal arm is inverse-quadratic in its "gap", times a factor which is logarithmic in many other parameters: the number of arms, probability of failure, and ratio $\lambda / \varepsilon$ which essentially quantifies the number of length-$\varepsilon$ subintervals that must be searched in the support of the arm's reward distribution to locate the $q$-quantile. Lower bounds are presented which show that the inverse-quadratic dependence on gap, logarithmic dependence on inverse failure probability, and logarithmic dependence on $\lambda/\varepsilon$ cannot be avoided.

The problem addressed in the paper is novel, and the algorithm and analysis are close to the best possible: the bounds are within logarithmic factors of the best possible, and some (though not all of the logarithmic factors) are shown to be unavoidable. I found the writing to be fairly clear, and the paper does a good job justifying assumptions (such as the relaxation of BAI to $\varepsilon$-BAI) when necessary. As one reviewer noted, the algorithm is not very innovative: it uses active arm elimination, combined with noisy binary search to detect when the elimination condition is met. As noted by another reviewer, the constant factors in the bounds are probably far from the best possible. I think both of these are fair critiques, but in my opinion the contribution is above the bar for ALT even in light of the critiques. A third critique that I found less persuasive is that the algorithm is only shown to be near-optimal in the threshold-query model, which allows for the possibility that algorithms making use of one-bit feedback in a different way might be able to improve the bounds significantly. I find this critique less persuasive than the others because in my view, the threshold-query model is important enough that it is valuable to study bounds for BAI in this model regardless of whether those bounds are also optimal in the more general setting of one-bit feedback.

One of the reviews identified some relevant literature on quantile estimation using threshold queries, and the authors agreed to revise the paper to include some discussion of these prior works. I strongly encourage them to do so.

**Paper Award:**

No